# Using the Path of Least Resistance to Explain Deep Networks

## Abstract

Integrated Gradients (IG), a widely used axiomatic path-based attribution method, assigns importance scores to input features by integrating model gradients along a straight path from a baseline to the input. While effective in some cases, we show that straight paths can lead to flawed attributions. In this paper, we identify the cause of these misattributions and propose an alternative approach that equips the input space with a model-induced Riemannian metric (derived from the explained model's Jacobian) and computes attributions by integrating gradients along geodesics under this metric. We call this method *Geodesic Integrated Gradients* (GIG). To approximate geodesic paths, we introduce two techniques: a $k$-Nearest Neighbours-based approach for smaller models and a Stochastic Variational Inference-based method for larger ones. Additionally, we propose a new axiom, No-Cancellation Completeness (NCC), which strengthens completeness by ruling out feature-wise cancellation. We prove that, for path-based attributions under the model-induced metric, NCC holds if and only if the integration path is a geodesic. Through experiments on both synthetic and real-world image classification data, we provide empirical evidence supporting our theoretical analysis and showing that GIG produces more faithful attributions than existing methods, including IG, on the benchmarks considered.

## 1 Introduction

The use of deep learning models has risen in many applications. With it, so too has the desire to understand why these models make certain predictions. These models are often referred to as "opaque", as it is difficult to discern the reasoning behind their predictions (Marcus, 2018). Additionally, deep learning models can inadvertently learn and perpetuate biases found in their training data (Sap et al., 2019). To create fair and trustworthy algorithms, it is essential to be able to explain a model's output (Das and Rad, 2020).

A variety of methods have been proposed to explain neural network predictions, each addressing different aspects of interpretability. Among them, path-based attribution methods have received particular attention due to their strong axiomatic foundations. Integrated Gradients (IG) (Sundararajan et al., 2017) is perhaps the most widely used such method, integrating model gradients along a straight path from a reference baseline to the input. Subsequent work has sought to improve IG's attributions by modifying the integration path: Guided Integrated Gradients (Kapishnikov et al., 2021) adapts paths with noise-reducing heuristics, Integrated Decision Gradients (Walker et al., 2024) emphasises regions where the model output changes most, and diffusion-based paths (Lei et al., 2024) keep interpolants near the training distribution. Complementary to path-based methods, perturbation-based approaches such as SHAP (Lundberg and Lee, 2017) and LIME (Ribeiro et al., 2016) explain predictions by observing how outputs change when inputs are perturbed, offering model-agnostic explanations at the cost of different theoretical trade-offs. We position our contribution relative to these approaches in Section 4.

Significant effort has been dedicated to designing explanation methods that satisfy certain desirable axioms. This is due to the lack of ground truth for evaluating them. The axioms can ensure that the explanations are principled. One of the most successful axiomatic methods is Integrated Gradients (IG) (Sundararajan et al., 2017). Consider a function $f : R^n \to R$, representing the neural network and an input vector $\mathbf{x} \in R^n$.

Original Image

Integrated Gradients

Geodesic Integrated Gradients

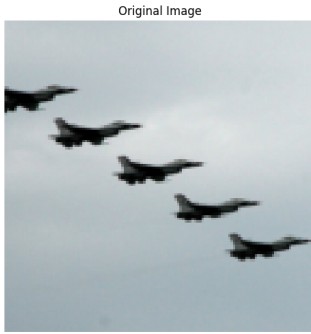 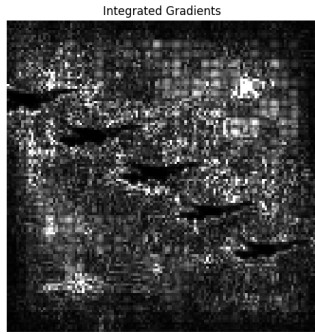 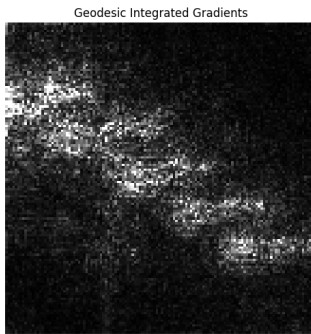

Figure 1: Comparison of attributions generated by Integrated Gradients (middle figure) and Geodesic Integrated Gradients (right figure) for image classification with a ConvNext model. Integrated Gradients follow straight paths in Euclidean space, which can result in misleading attributions. In contrast, Geodesic Integrated Gradients integrate along geodesic paths on a Riemannian manifold defined by the model, correcting misattributions caused by poor alignment with the model's gradient landscape. In both cases, the baseline is a black image. For IG, since the jets are black, apart from the artefacts created outside of the boundaries of the jets, the attribution method is misled into considering the jets unimportant for classification, even though they are the objects being classified. Geodesic IG does not suffer from this issue. Further examples of such misattributions due to black segments in images are shown in Appendix B.

Furthermore, consider a baseline input vector $\overline{\mathbf{x}} \in R^n$ (typically chosen such that the network gives baseline a near zero score). IG produces an attribution for the network's prediction by quantifying how much of the difference $f(\mathbf{x}) - f(\overline{\mathbf{x}})$ can be attributed to the $i$th dimension of $\mathbf{x}$, $\mathbf{x}_i$. In other words, for each input feature, IG assigns a score reflecting that feature's contribution to the change in model output between the baseline and the input.

Integrated Gradient gives attribution $IG_i$ to the $i$th dimension of the input by approximating the following path integral

$$IG_i(\mathbf{x}) = (\mathbf{x}_i - \overline{\mathbf{x}}_i) \int_0^1 \frac{\partial f(\gamma(t))}{\partial \mathbf{x}_i} dt, \tag{1}$$

where $\gamma(t) = \overline{\mathbf{x}} + t(\mathbf{x} - \overline{\mathbf{x}})$ is a straight path from the baseline to input. The claim of the creators of IG is that Eq. 1 tells us how the model got from predicting essentially nothing at $\overline{\mathbf{x}}$ to giving the prediction at $\mathbf{x}$. Considering gradients represent the rate of change of functions, the above expression should tell us how scaling each feature along the path affects the increase in the network score for the predicted class.

**In this paper**, we demonstrate that defining attributions along straight paths in Euclidean space can lead to flawed attributions. We examine the consequences of these misattributions through examples in computer vision, as shown in Fig. 1, along with simpler, illustrative cases in Fig. 3. To address these issues, we introduce **Geodesic Integrated Gradients**, a generalisation of IG that replaces straight paths with geodesic ones. These geodesics are defined on a Riemannian manifold, characterised by the model's input space and a metric tensor induced by the model's gradients. This approach mitigates the identified pitfalls while retaining all the axioms of IG. Furthermore, we introduce an additional axiom, No-Cancellation Completeness (NCC), which strengthens the completeness axiom by ruling out cancellation between feature attributions. We prove that Geodesic Integrated Gradients is the unique path-based method that satisfies this axiom, further justifying its theoretical soundness.

Before making the case for our Geodesic Integrated Gradient, let us first show an example of an artefact that can arise from choosing straight paths, generating explanations which do not reflect the true behaviour of a model.

We highlight this issue with a simple half-moons classification task. We train a three-layer multi-layer perceptron (MLP) with ReLU activations and a cross-entropy loss to distinguish the upper moon from the lower one. The cross-entropy is decomposed into a final log-softmax activation followed by a negative log-likelihood loss, allowing us to explain probabilities. The model is trained long enough that high gradients emerge at the decision boundary, while the model remains flat elsewhere, as illustrated by the gradient contour maps in Fig. 3.

We now compute Integrated Gradients, Eq. 1, for this model on the test data. As an appropriate baseline, we choose the point $(-0.5, -0.5)$, which is a reasonable choice since the network should assign a near-zero score to it. The input $\mathbf{x}$ has two features: $x_0$ (horizontal axis) and $x_1$ (vertical axis). In Fig. 3(a) (top image), we colour each input point according to its attribution value for the horizontal feature, $\mathrm{IG}_0(\mathbf{x})$, which is the importance score assigned to the horizontal coordinate of that point.

In this half-moons experiment, the trained model is empirically nearly constant away from the decision boundary (as shown by the output/gradient contours in Fig. 3). This is a common property of well-trained classifiers with saturating outputs (here, the log-softmax): once the model is confident in its prediction, the output plateaus and gradients become negligible. Therefore, for points sufficiently far from the boundary, small displacements further into the region change $f(x)$ negligibly, and we expect attributions to be stable across those regions. More precisely, for two points $\mathbf{x}_1, \mathbf{x}_2$ in the same flat region with identical baselines, a faithful attribution method should assign similar total attribution magnitudes, since $f(\mathbf{x}_1) \approx f(\mathbf{x}_2)$. Yet, as shown in Fig. 3(a), Integrated Gradients significantly violates this for certain points. These are points, as seen in the figure, where the straight-line path from the baseline $(-0.5, -0.5)$ to the input passes mostly through high-gradient regions. This does not accurately reflect the model's behaviour. A similar issue arises in the vertical attribution, shown in Fig. 3(b), but in the opposite direction to the horizontal attribution.

In contrast, Geodesic Integrated Gradients (Geodesic IG), shown at the bottom of Fig. 3(a) and (b), correctly assigns equally high attributions to all points sufficiently far from the decision boundary. In Section 3.1, we detail our method and explain how it achieves the results presented in this figure.

The above artefacts further highlight an issue with over-reliance on the following axiom satisfied by IG:

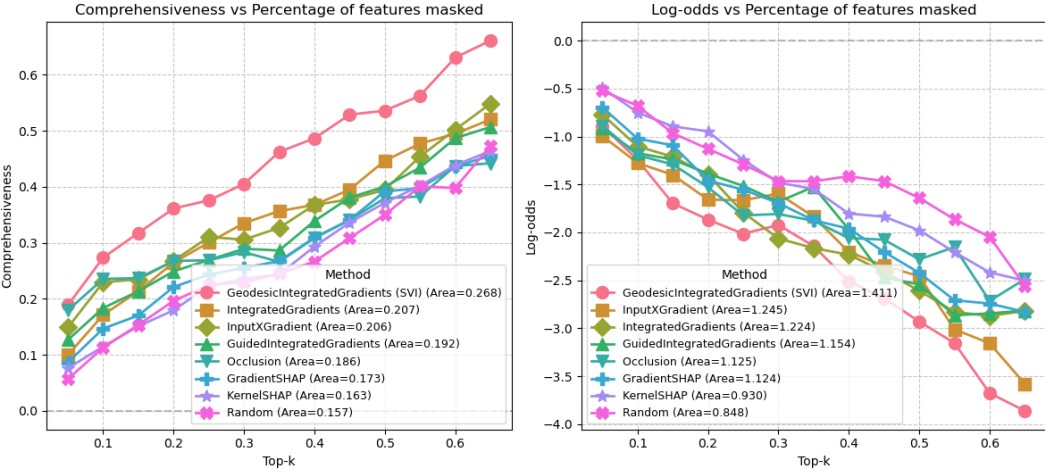

Figure 2: **Faithfulness metrics used in the Pascal VOC experiment with ConvNext.** For each attribution method, we rank pixels by $|A_i(\mathbf{x})|$ and mask the top $k\%$. Comprehensiveness measures the drop in predicted probability of the target class after masking (higher is better). Log-odds measures the log-odds after masking (more negative indicates stronger removal; to keep 'higher is better' we report an 'area-over-curve'). Curves are summarised by area-based aggregates (AUC/AOC). Geodesic IG significantly outperforms other methods in both metrics. See Section 3 for details of the experiments.

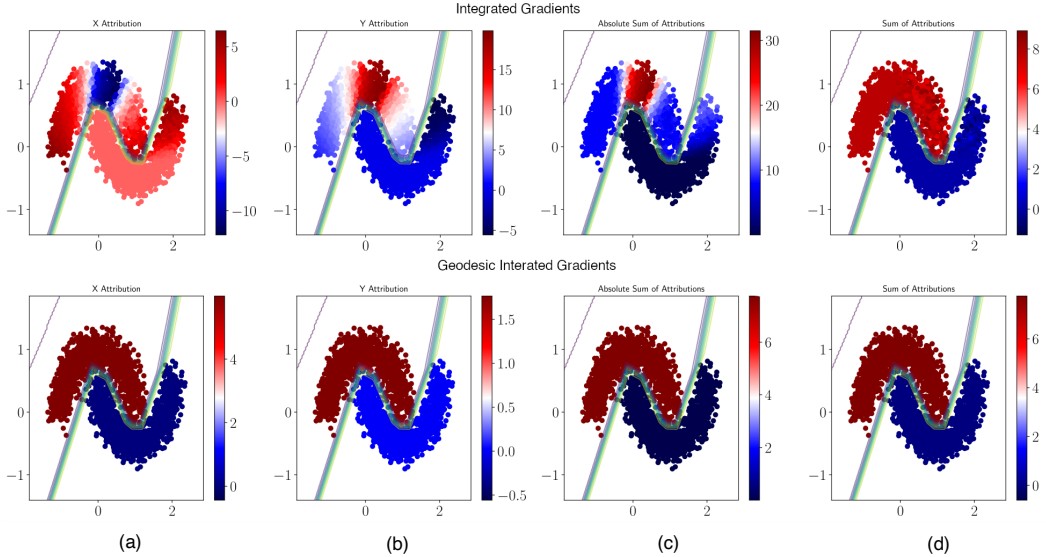

Figure 3: **Integrated Gradients (IG) attributions (top) vs. Geodesic IG (bottom).** We plot scatter plots of 10,000 samples from the half-moons dataset with noise parameter $\mathcal{N}(0, 0.15)$. An MLP model is trained for classification, and the model gradients are shown as contour maps. The model is nearly flat everywhere except at the decision boundary. Using a baseline at $(-0.5, -0.5)$, we compute IG and Geodesic IG attributions. From left to right, the colour maps display (a) the attribution score $A_0(\mathbf{x})$ for the horizontal-axis feature $x_0$, (b) the attribution score $A_1(\mathbf{x})$ for the vertical-axis feature $x_1$, (c) the total absolute attribution $\sum_i |A_i(\mathbf{x})|$, and (d) the total signed attribution $\sum_i A_i(\mathbf{x})$. According to Axioms 1 and 2, the heatmaps in the last two columns should resemble those in Fig. 4. As shown, IG satisfies Axiom 1 (last column) but not Axiom 2 (penultimate column). In contrast, Geodesic IG satisfies both. Additionally, similar to Fig. 1, IG is highly sensitive to the choice of baseline due to its reliance on a straight-line path, whereas Geodesic IG reduces this sensitivity by adapting to the model's gradient landscape.

**Axiom 1 (Completeness)** *Consider an input-baseline pair $\boldsymbol{x}$ and $\overline{\boldsymbol{x}}$, and a function $f$. Let $A_i(\boldsymbol{x})$ be the attribution of $\boldsymbol{x}_i$. The method satisfies Completeness if*

$$\sum_i A_i(\boldsymbol{x}) = f(\boldsymbol{x}) - f(\overline{\boldsymbol{x}}), \tag{2}$$

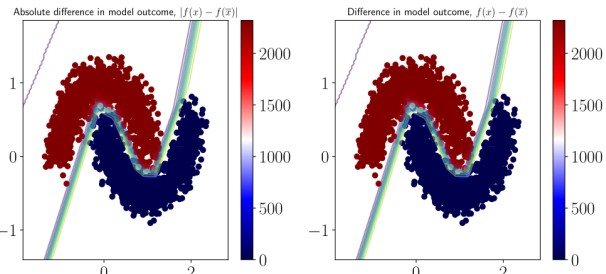

Figure 4: **Model output at the baseline vs. input points.** To assess whether our attribution methods satisfy Axioms 1 and 2 in the half-moons example, we plot the model output at the input points, subtracting the model output at the baseline. The left plot shows this difference, $f(\mathbf{x}) - f(\overline{\mathbf{x}})$, while the right plot shows the absolute difference, $|f(\mathbf{x}) - f(\overline{\mathbf{x}})|$. Comparing these plots with those in Fig. 3, we observe that Geodesic IG satisfies both axioms, whereas IG satisfies Completeness, only.

This axiom ensures that the total attribution equals the difference in function values between the baseline and the input. While completeness guarantees a basic additivity property, that the attributions sum to the baseline-to-input change in model output, Eq. (2), it does not prevent cancellation between features. Specifically, completeness ensures only that the *signed sum* of attributions equals $f(\mathbf{x}) - f(\overline{\mathbf{x}})$, but individual attribution values may be arbitrarily inflated or deflated as long as they sum correctly.

In the above example, we considered a two-dimensional case where the model assigns nearly the same function value to the input points belonging to the top half-moon. Completeness then requires that for all points, the total attribution must be close to the same constant, $c$, i.e. $\sum_i A_i(\mathbf{x}) \approx c \; \forall \mathbf{x}$. Furthermore, we *know* that the horizontal feature of all points in the same half-moon have equal importance for this particular model. The same goes for the vertical features. Despite this, while IG violates the latter, it can still satisfy completeness by assigning large positive attributions to one feature and equally large negative attributions to the other, cancelling out in the sum but distorting individual attributions. This imbalance does not occur consistently across all points, further complicating interpretation.

To address this issue, we introduce a stronger axiom:

**Axiom 2 (No-Cancellation Completeness (NCC))** *Consider an input-baseline pair $\boldsymbol{x}$ and $\overline{\boldsymbol{x}}$, and a continuously differentiable function $f$. An attribution method satisfies NCC if*

$$\sum_i |A_i(\boldsymbol{x})| = |f(\boldsymbol{x}) - f(\overline{\boldsymbol{x}})| \tag{3}$$

As we shall see in section 2.5.1, Geodesic Integrated Gradients is the only path-based method that satisfies this desired axiom.

**Why NCC in addition to completeness?** For path-integral attributions of the form in Eq. 12, standard completeness $\sum_i A_i(x) = f(x) - f(x')$ holds automatically by the fundamental theorem of calculus whenever the path endpoints are fixed. Completeness alone, however, permits cancellation: a method can assign a large positive attribution to one feature and a large negative attribution to another, making the sum correct while making individual feature scores misleading. NCC prevents this failure mode by requiring that attribution mass does not cancel across features. To see why this matters, consider the analogy of a financial ledger: completeness requires that all entries sum to the correct total (a balance sheet that balances), while NCC additionally requires that there are no hidden offsetting transactions. When attributions cancel, the total $\sum_i |A_i(\mathbf{x})|$ exceeds $|f(\mathbf{x}) - f(\overline{\mathbf{x}})|$, indicating that the method attributes more importance than the model output change warrants.

**Relationship between NCC and completeness.** We emphasise that NCC is not a strict strengthening of completeness in the logical sense: satisfying NCC does not automatically imply satisfying completeness, nor vice versa. However, when both axioms are satisfied simultaneously, as is the case for Geodesic IG (Theorem 1 establishes NCC, while completeness follows from the fundamental theorem of calculus for any path-based method with fixed endpoints), the attributions are constrained from both sides: they sum to the correct total *and* no cancellation occurs between features. We chose the name "No-Cancellation Completeness" to describe the axiom's content (completeness of absolute attributions) rather than to imply it logically subsumes standard completeness.

In Section 2, we first provide background on notation and evaluation metrics (Section 2.1), then present two methods for approximating the geodesic path between two points on a manifold. The first method, based on $k$-nearest neighbours ($k$NN), is designed for low-dimensional inputs, while the second method, utilising Stochastic Variational Inference, is suited for high-dimensional inputs such as images. We further demonstrate that Geodesic IG adheres to all the axioms of Integrated Gradients.

In Section 3, we demonstrate the effectiveness of the Geodesic IG method on the real-world Pascal VOC 2012 dataset (Everingham et al.). Our results outperform existing methods, as we evaluate using two metrics. We preview the results of this experiment in Fig. 2.

Section 4 reviews related work, including the comparison of Geodesic IG with other methods that attempt to overcome the shortcomings of Integrated Gradients.

## 2 Method

### 2.1 Background and notation

We briefly consolidate key definitions and notation used throughout the paper.

**Setup.** We consider a neural network $f : \mathbb{R}^n \to \mathbb{R}$ mapping an $n$-dimensional input $\mathbf{x}$ to a scalar output (e.g., the log-probability of a target class; see Remark 3 for multi-class extension). A *baseline* $\overline{\mathbf{x}} \in \mathbb{R}^n$ represents an uninformative reference input (e.g., a black image) to which the model assigns a near-zero score. The goal of attribution is to assign each feature $x_i$ a score $A_i(\mathbf{x})$ reflecting its contribution to $f(\mathbf{x}) - f(\overline{\mathbf{x}})$.

**Integrated Gradients.** IG (Sundararajan et al., 2017) computes attributions by integrating the model's partial derivatives along a straight path $\gamma(t) = \overline{\mathbf{x}} + t(\mathbf{x} - \overline{\mathbf{x}})$ from baseline to input, as defined in Eq. (1). The method satisfies several desirable axioms: *Completeness* (Axiom 1), which ensures attributions sum to $f(\mathbf{x}) - f(\overline{\mathbf{x}})$; *Sensitivity*, which ensures that if the model depends on a feature, that feature receives non-zero attribution; and *Implementation Invariance*, which ensures attributions depend only on the function $f$, not on its internal implementation. As shown in Sundararajan et al. (2017), all path-based generalisations of IG (replacing the straight path with an arbitrary path $\gamma$) satisfy these axioms except possibly *Symmetry*.

**Evaluation metrics.** We evaluate attribution quality using standard faithfulness metrics. *Comprehensiveness* (DeYoung et al., 2019) measures the drop in predicted probability when the most important features are masked: higher values indicate that the identified features are indeed important. *Log-odds* (Shrikumar et al., 2017) measures the log-odds of the target class after masking: more negative values indicate stronger evidence removal. We also introduce *Purity* for the half-moons setting (Eq. 14), which measures whether high-attribution points belong to the correct class. All metrics are summarised across multiple thresholds or noise levels via area-based aggregates (AUC or AOC), where higher values are better for all reported summaries (see Sections 3.1 and 3.2 for precise definitions).

### 2.2 Geodesic distance formulation

In Section 1, we gave the intuition that using geodesic paths can correct the misattributions in IG that arise from integrating along straight paths. Let us now formalise this idea. After that, we introduce two approximation methods in Sections 2.3 and 2.4. Both target the same object: a geodesic under the model-induced metric Eq. (5), i.e., a path that minimises accumulated local "resistance" induced by the model gradient geometry. They differ only in how the path search is operationalised:

- *$k$NN shortest-path* (Section 2.3) discretises the space using a sampled point set and reduces geodesic finding to a weighted shortest-path problem; it is most appropriate when the input dimensionality is low enough (e.g., $n = 2$) that a few thousand samples can densely cover the region between baseline and input.

- **Energy-based variational paths** (Section 2.4) optimise a continuous path parameterisation to avoid high-gradient regions without requiring dense global sampling; it is most appropriate in high-dimensional domains (e.g., images with $n \gg 100$) where graph construction would require prohibitively many samples.

A simple rule of thumb: if the input space can be reasonably covered by a few thousand samples (low-dimensional tabular or synthetic data), use $k$NN; for typical deep learning inputs (images, audio, embeddings), use the energy-based method.

Let us define a neural network as a function $f : \mathbb{R}^n \to \mathbb{R}$, where $n$ is the dimension of the input space. Let us also define $\mathbf{x}$ a point in this input space. We denote the Jacobian of $f$ at $\mathbf{x}$ as $\mathrm{J}_{\mathbf{x}}$. Using Taylor's theorem, for a vector $\boldsymbol{\delta}$ with an infinitesimal norm: $\forall \epsilon, ||\boldsymbol{\delta}|| \leq \epsilon$, we have:

$$||f(\mathbf{x} + \boldsymbol{\delta}) - f(\mathbf{x})|| \approx ||\mathrm{J}_{\mathbf{x}}\boldsymbol{\delta}|| \approx \boldsymbol{\delta}^T \mathrm{J}_{\mathbf{x}}^T \mathrm{J}_{\mathbf{x}} \boldsymbol{\delta} \tag{4}$$

Using equation 4, we can now define a tangent space $T_{\mathbf{x}}M$ of all $\boldsymbol{\delta}$, equipped with a local inner product $G_{\mathbf{x}}$:

$$< \boldsymbol{\delta}, \boldsymbol{\delta'} >_{\mathbf{x}} = \boldsymbol{\delta}^T G_{\mathbf{x}} \boldsymbol{\delta'} = \boldsymbol{\delta}^T J_{\mathbf{x}}^T J_{\mathbf{x}} \boldsymbol{\delta'} \tag{5}$$

As a result, we can view the input space as a Riemannian manifold $(\mathbb{R}^n, G)$, where the Riemannian metric G is defined above. On this manifold, the length of a curve $\gamma(t) : [0,1] \to \mathbb{R}^n$ is defined as:

$$\begin{aligned} L(\gamma) &= \int_0^1 \sqrt{< \dot{\gamma}(t), \dot{\gamma}(t) >_{\gamma(t)}} dt \\ &= \int_0^1 ||\partial_t f(\gamma(t)) \times \dot{\gamma}(t)|| \, dt, \end{aligned} \tag{6}$$

where $\dot{\gamma}(t)$ is the derivative of $\gamma(t)$ with respect to $t$. The **geodesic distance**, denoted $L^*$, between **a** and **b** is then defined as the minimum length among curves $\gamma$ such that $\gamma(0) = \mathbf{a}$ and $\gamma(1) = \mathbf{b}$. We also call **geodesic path** the curve $\gamma^*$ which minimises the length L. This path can be interpreted as the shortest path between **a** and **b** in the manifold.

**Remark 1** *We can infer from Equation 6 that the geodesic path avoids as much as possible high-gradients regions. This is the main desired property of a path to be used for path-based attributions. Representing the path of least resistance, the geodesic path circumvents superficially high values of attributions.*

**Remark 2 (Metric degeneracy and regularisation)** *The metric $G_{\boldsymbol{x}} = J_{\boldsymbol{x}}^T J_{\boldsymbol{x}}$ is rank-1 for scalar-valued functions $f : \mathbb{R}^n \to \mathbb{R}$, which means it is technically degenerate (semi-Riemannian rather than Riemannian). In regions where $\nabla f(\boldsymbol{x}) = 0$ (e.g., at local optima or in flat regions), the metric vanishes entirely, and geodesics under this metric are not uniquely defined. However, this degeneracy is precisely what we exploit: in flat regions, the "cost" of traversing any direction is zero, so the path can take any route without penalty. Our approximation methods handle this naturally: the kNN method assigns zero weight to edges in flat regions (allowing any path), while the SVI method's distance penalty keeps paths reasonably direct when gradient-based costs vanish. In practice, for well-trained classification models, the gradient is non-zero along most of the path from baseline to input, particularly near decision boundaries where attribution matters most.*

*A natural alternative is to regularise the metric as $G_{\boldsymbol{x}}^{(\varepsilon)} = J_{\boldsymbol{x}}^T J_{\boldsymbol{x}} + \varepsilon \boldsymbol{I}$, which yields a proper (non-degenerate) Riemannian metric for any $\varepsilon > 0$. In the limit $\varepsilon \to 0$, geodesics under $G^{(\varepsilon)}$ converge to those under our degenerate metric (where they exist), and for $\varepsilon \to \infty$ geodesics converge to Euclidean straight lines, recovering standard IG. Our theoretical results (Theorem 1) apply in the idealised $\varepsilon = 0$ setting; we expect them to hold approximately for small $\varepsilon$, though a formal perturbation analysis is left for future work. We note that the connection between $J^T J$ and the Fisher Information Metric in information geometry (Dombrowski et al., 2019) provides additional theoretical grounding for this choice of metric.*

**Remark 3 (Extension to multi-class classification)** *For multi-class classification, we explain the probability of a single target class by applying a log-softmax transformation and treating the resulting scalar as $f$. This is the standard approach used in IG and related methods. The metric is then induced by the gradient of this scalar output with respect to inputs.*

### 2.3 Approximation of the geodesic with $K$ Nearest Neighbours.

Computing the exact geodesic would require computing $L$ on an infinite number of paths $\gamma$, which is not possible in practice. However, several methods have been proposed to approximate this value. We draw from previous work (Yang et al., 2018; Chen et al., 2019) and present one with desirable characteristics.

First, we compute the $K$ Nearest Neighbours ($k$NN) algorithm on points between (and including) input and baseline. These points can be either sampled or generated. The geodesic distance between two neighbouring points, $\mathbf{x}_i$ and $\mathbf{x}_j$, can be approximated by a straight path $\mathbf{x}_i + t \times (\mathbf{x}_j - \mathbf{x}_i)$. We have the above approximation because for dense enough data, the euclidean distance between neighbouring points is a good

approximation of the geodesic distance. This reflects the fact that a small region of a Riemannian manifold, called Riemann neighbourhood, is locally isometric to a Euclidean space[1]. So the geodesic distance between the two neighbouring points is approximated by:

$$
\begin{aligned}
\mathrm{L}_{ij}^* &= \int_0^1 ||\partial_t f(\mathbf{x}_i + t \times (\mathbf{x}_j - \mathbf{x}_i)) \times (\mathbf{x}_i - \mathbf{x}_j)|| \, dt \\
&= ||\mathbf{x}_i - \mathbf{x}_j|| \int_0^1 ||\partial_t f(\mathbf{x}_i + t \times (\mathbf{x}_j - \mathbf{x}_i))|| \, dt
\end{aligned}
\tag{7}
$$

Equation 7 corresponds to the original Integrated Gradients method, albeit with the norm. This integral can be approximated by a Riemann sum similarly to Sundararajan et al. (2017):

$$
\mathrm{L}_{ij}^* \approx ||\mathbf{x}_i - \mathbf{x}_j|| \sum_{k=0}^m ||\partial f(\mathbf{x}_i + \frac{k}{m} \times (\mathbf{x}_j - \mathbf{x}_i))||
\tag{8}
$$

For input-baseline pair, $\mathbf{x}$ and $\overline{\mathbf{x}}$, we can now construct a weighted graph from the sampled points as follows. Concretely, we build a graph $G = (V, E)$ where each sampled point (including the input $\mathbf{x}$ and baseline $\overline{\mathbf{x}}$) is a node in $V$. For each node $\mathbf{x}_i$, we add undirected edges to its $k$ nearest neighbours under Euclidean distance; these edges form $E$. Each edge $(i, j) \in E$ is assigned a weight $w_{ij}$ equal to our approximation of the local geodesic length between $\mathbf{x}_i$ and $\mathbf{x}_j$, as given by Eq. (8). Intuitively, edges passing through high-gradient regions receive large weights (since the integral of gradient norms is high along them), making them "expensive" to traverse. Edges in flat regions receive small weights. Finding a geodesic from baseline to input is then reduced to a standard shortest-path problem on this weighted graph, from node $\overline{\mathbf{x}}$ to node $\mathbf{x}$. The resulting node sequence defines a piecewise-linear path along which we compute path-integrated gradients. This approach is analogous to the Isomap algorithm (Tenenbaum et al., 2000), which approximates geodesic distances on data manifolds using shortest paths in neighbourhood graphs. To compute the geodesic path between $\mathbf{x}$ and $\overline{\mathbf{x}}$, we can use a shortest path algorithm, such as Dijkstra or $\mathrm{A}^*$ with the euclidean distance as the heuristic.

The resulting Geodesic Integrated Gradients corresponds to the sum of the gradients along this shortest path:

$$
\begin{aligned}
&\text{Geodesic IG}_i(\mathbf{x}) = \\
&(x_i - \overline{x}_i) \sum_{k=0}^m \int_0^1 \frac{\partial f(\mathbf{x}^k + t \times (\mathbf{x}^{k+1} - \mathbf{x}^k))}{\partial x_i} \, dt
\end{aligned}
\tag{9}
$$

where $\mathbf{x}^k$ are the points along the shortest path. The integrals in Equation 9 can also be approximated with Riemann sums.

The gradients between each pair of neighbours can also be estimated in batches to speed up the attribution computation. Moreover, several inputs' attributions can be computed together, with similar speed as IG: if we want to compute the attribution of $N$ inputs, with 10 interpolation steps and 5 nearest neighbours, the number of gradients to calculate is $10 \times 5 \times N = 50N$, which amounts to computing IG with 50 steps. This does not include the computation of the shortest path, which is for instance $O(N^2)$ for Dijkstra algorithm. See Fig. 5 for an illustration of this method.

**Assumption of the approximation.** Here we formalise the intuition that, for a pair of neighbours, the geodesic path between them is close to the euclidean one. Notice that the derivative of the neural network $f$ is Lipschitz continuous,

---

[1]We shall further formalise this intuition later in this section.

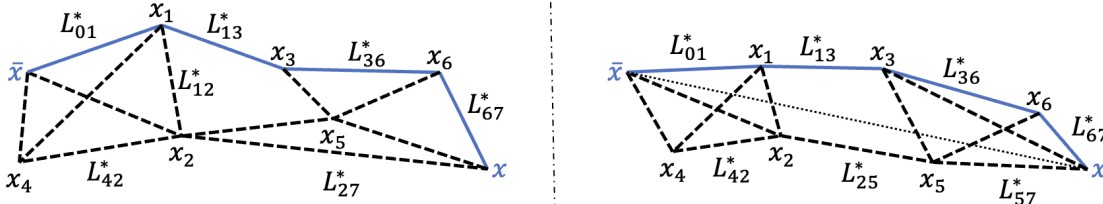

Figure 5: **Method overview.** For an input $\mathbf{x}$, a baseline $\bar{\mathbf{x}}$, and a set of points $\mathbf{x}_i$, we compute the $k$NN graph using the euclidean distance (dashed lines). For each couple $(\mathbf{x}_i, \mathbf{x}_j)$, we then compute the integrated gradients $\mathrm{L}_{ij}^*$ using Equation 8. For clarity, not all $\mathrm{L}_{ij}^*$ are present on the figure. Using the resulting undirected weighted graph, we use the Dijkstra algorithm to find the shortest path between $\mathbf{x}$ and $\bar{\mathbf{x}}$ (blue continuous lines). On the left, the points $\mathbf{x}_i$ are provided while, on the right, the points are generated along the straight line between $\mathbf{x}$ and $\bar{\mathbf{x}}$ (dotted line).

$$\exists \mathrm{K} \, \forall \mathbf{x}, \mathbf{y}, \, ||\mathrm{J}_\mathbf{x} - \mathrm{J}_\mathbf{y}|| \leq \mathrm{K} \times ||\mathbf{x} - \mathbf{y}||. \tag{10}$$

Equation 10 is equivalent to the Hessian of $f$ being bounded. Under this assumption, if two points $\mathbf{x}$ and $\mathbf{y}$ are close enough, the Jacobian of one point is approximately equal to the other: if $||\mathbf{x} - \mathbf{y}|| \leq \epsilon$, then $\mathrm{J}_\mathbf{x} \approx \mathrm{J}_\mathbf{y}$. As a result, the length between $\mathbf{x}$ and $\mathbf{y}$, for a curve $\gamma$, is: $\mathrm{L}(\gamma) \approx \int_\gamma ||\mathrm{J}_\mathbf{x}|| \, d\mathbf{x} \approx ||\mathrm{J}_\mathbf{x}|| \int_\gamma d\mathbf{x}$. Due to the triangular inequality, the shortest path $\gamma^*$ is then a straight line, and we have: $\mathrm{L}^*(\mathbf{x}, \mathbf{y}) \approx ||\mathrm{J}_\mathbf{x}|| \times ||\mathbf{x} - \mathbf{y}||$.

As a result, under this assumption, if two points are close, the geodesic path can be approximated with a straight line. Note that even though we take the path between two neighbouring points to be a straight line, we do not assume that the Jacobian of the function between the two points is constant.

**Handling disconnected graphs** An issue with the graph computed with the $k$NN algorithm is that it could be disconnected, in which case it could be impossible to compute a path between an input and a baseline. Disconnection is most common when k is small relative to the intrinsic complexity of the sampled set (or when the sampled set is sparse in regions needed to connect baseline to input), so there is no chain of neighbour relations linking the two endpoints. To alleviate this issue, we add so-called "bridges" to the graph, as follows: for each disconnected component, we add one link between them, specifically between two points of each component having the lowest euclidean distance. An illustration of this method is displayed on Figure 6.

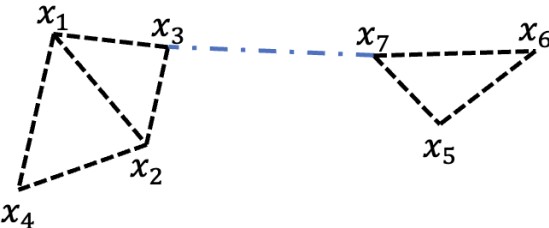

Figure 6: When the $k$NN graph is disconnected, as illustrated here, it would be impossible to compute Geodesic IG between certain points, for instance $\mathbf{x}_1$ and $\mathbf{x}_5$ here. To solve this, we add a single link between disconnected graphs, here between $\mathbf{x}_3$ and $\mathbf{x}_7$.

However, we stress that this solution is not optimal, and argue that a better way of handling this issue would be to avoid disconnected graphs in the first place. This can be done by increasing the number of neighbours $k$. In high-dimensional settings (e.g., images), ensuring connectivity typically requires prohibitively dense

sampling, which motivates the alternative energy-based approximation in Section 2.4. In practice, we found that for the half-moons experiments ($n = 2$) with 10,000 samples and $k = 15$, the graph was always connected, so bridging was not required. The bridging mechanism is a safety net for edge cases (e.g., unusual baseline locations or sparse regions) rather than a routine requirement. When bridging is needed, the introduced edge has high geodesic weight (since the connected components are far apart), so the shortest path will use it only as a last resort, limiting its impact on attribution quality.

**Approximation quality.** Our $k$NN-based geodesic approximation relies on two sources of error: (1) the discretisation error from approximating a continuous geodesic with a piecewise-linear path through sampled points, and (2) the local approximation error from treating short segments as straight lines. Under the Lipschitz assumption (Eq. 10), the local error scales as $O(\epsilon^2)$ where $\epsilon$ is the maximum edge length. The global approximation quality depends on the sampling density relative to the curvature of the true geodesic. While we do not provide formal approximation bounds, our experimental results suggest that the approximation is sufficient to capture the essential behaviour of avoiding high-gradient regions. Developing tighter theoretical guarantees is an interesting direction for future work.

### 2.4 Approximation of the geodesic with energy-based sampling.

While our $k$NN-based method is effective for explaining simpler manifolds (typically low-dimensional synthetic/tabular inputs, or low-dimensional learned embeddings), its applicability diminishes as model complexity increases. In such cases, a prohibitively large number of samples is required between the baseline and the input to provide accurate estimates of the geodesic path. Even with relatively large number of samples, it is not trivial where on the manifold to sample the points to adequately capture the gradient landscape. Furthermore, once the points are sampled, searching the graph for the shortest path will be computationally too intensive. For such use-cases, in this subsection, we devise an energy-based sampling method as another approximation procedure.

As noted in Remark 1, we aim to sample from the shortest paths between two points in the input space while avoiding regions of high gradients. To achieve this, we deviate from the straight line to minimise the influence of high-gradient areas. This process can be approximated as follows: we begin with a straight-line path between the two points and define a potential energy function composed of two terms: a distance term to maintain proximity to the straight line and a curvature penalty term to push away from high gradient regions. Minimising this energy function approximates the geodesic path.

Formally, the distance term is defined as $d(\mathbf{x}, \mathbf{y}) := \|\mathbf{x} - \mathbf{y}\|_2$, and the curvature term as $c(\mathbf{x}) := \|\nabla f(\mathbf{x})\|_2$ where $f$ represents the neural network. The total energy being minimised is

$$E(\gamma) = \sum_{i=1}^{n} d(\gamma_i, \gamma_i^0) - \beta c(\gamma_i), \tag{11}$$

where $\gamma$ is the path, $\gamma^0$ is the initial path, and $\beta$ controls the trade-off between distance and curvature.

**Relationship to true geodesics.** We emphasise that minimising $E(\gamma)$ in Eq. 11 is *not* equivalent to minimising the Riemannian length functional (Eq. 6), so the resulting paths are heuristic approximations to geodesics rather than exact solutions of the geodesic equation. The energy trades off proximity to the straight line against gradient avoidance, which captures the essential behaviour of geodesics under our metric, avoiding high-gradient regions, without solving the geodesic ODE directly. The theoretical guarantees of Theorem 1 apply to true geodesics; for the SVI-approximated paths, the NCC axiom holds only approximately, with the approximation quality depending on how closely the energy-minimising path resembles the true geodesic. Despite this gap, our experiments demonstrate that even these approximate paths yield substantially better attributions than straight-line paths, suggesting that the key benefit, gradient avoidance, is captured by the heuristic.

With this energy function, one can use a suitable sampling method, such as Stochastic Variational Inference (SVI) or Hamiltonian Monte Carlo to sample points on the geodesic paths. Here we briefly describe the SVI optimisation, as this has a suitable balance of computational efficiency and accuracy.

SVI provides a probabilistic framework for optimising paths between input and baseline points. To achieve this, it defines a probability distribution $p(\gamma|\gamma_0)$ proportional to $exp(-E(\gamma))$, where $E(\gamma)$ is our defined potential energy. Rather than directly sampling from this complex distribution, we introduce a simpler variational distribution $q(\gamma)$ parametrised by learnable means and scales. This guide distribution takes the form of a factorised normal distribution $\prod_i N(\mu_i, \sigma_i)$ over path deviations.

The optimisation proceeds by minimising the KL divergence between $q(\gamma)$ and the true posterior through maximisation of the Evidence Lower Bound. Critically, this allows us to learn optimal parameters for $q(\gamma)$ through gradient-based optimisation. The learned means $\mu_i$ define the optimal path deviations from the initial straight-line path, while the scales $\sigma$ capture uncertainty in these deviations. This probabilistic approach naturally samples of the low-energy regions.

We apply this method in our computer vision experiments, demonstrating its efficacy in Section 3. However, for clarity, we visualise these paths on a simpler 2D half-moons example in Fig. 7. While the $k$NN method would typically be preferred for such simpler cases due to its ease of control, this example serves as an instructive illustration.

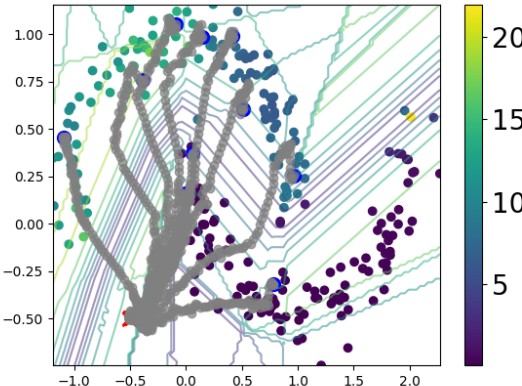

Figure 7: **Visualisation of 10 random paths.** For the simple case of half-moons with 1,000 samples, we display the sampled paths between 10 pairs of points. In low-gradient regions, the sampler favours straight lines, whereas in high-gradient regions, the paths adjust to become nearly perpendicular to the large gradient vectors, crossing these regions as quickly as possible.

Of course, using the SVI method comes with its own challenges. For example, in this case a suitable value of $\beta$ for the potential energy, as well as learning rate for the SVI algorithm itself needs to be chosen. As for any standard machine learning training, these values can be chosen using hyperparameter tuning, as we discuss in section 3.

**Hyperparameter guidance.** In practice, $\beta$ controls the trade-off between path directness (low $\beta$) and gradient avoidance (high $\beta$). We found that setting $\beta$ to be proportional to the average gradient magnitude along the straight-line path provides a reasonable initialisation. If $\beta$ is too low, the path collapses to the straight line (recovering standard IG); if too high, the path deviates excessively, increasing noise and potentially moving far from meaningful regions. For the SVI learning rate, standard practices apply: we used Adam with a learning rate of $10^{-2}$ and found results stable across $10^{-3}$ to $10^{-1}$. We acknowledge that a formal sensitivity analysis would be valuable, and we provide the values used in our experiments to facilitate reproducibility. Future work could develop adaptive schemes for $\beta$ that adjust based on the local gradient landscape.

## 2.5 Axiomatic properties

Designing an effective attribution method is challenging, partly because there are often no ground-truth explanations. If a method assigns importance to certain features in a counterintuitive way, it can be difficult to determine whether the issue lies with the model, the data, or the attribution method itself.

To address this, some attribution methods follow an axiomatic approach, defining principles that guide their design. Integrated Gradients is one such method, satisfying several key axioms. As shown in Sundararajan et al. (2017), all path-based methods (a generalisation of IG to arbitrary paths) uphold these axioms except for Symmetry.

Additionally, as discussed in Section 1, beyond IG's axioms, we introduce NCC (Axiom 2). Therefore, this subsection focuses on NCC and Symmetry only.

### 2.5.1 No-Cancellation Completeness

Here we prove that Geodesic IG satisfies Axiom 2 (NCC) and that, under the model-induced metric, NCC holds if and only if the integration path is geodesic.

**Theorem 1 (No-Cancellation Completeness)** *Let $f : \mathbb{R}^n \to \mathbb{R}$ be continuously differentiable with $\nabla f \neq 0$ along $\gamma$ (except possibly at isolated points), and let $\overline{\mathbf{x}}, \mathbf{x} \in \mathbb{R}^n$. Given a smooth path*

$$\gamma : [0, 1] \to \mathbb{R}^n, \quad \gamma(0) = \overline{\mathbf{x}}, \quad \gamma(1) = \mathbf{x},$$

*define its attributions as*

$$A_i^{\gamma}(\mathbf{x}) = \int_0^1 \frac{\partial f}{\partial x_i}(\gamma(t)) \, \dot{\gamma}_i(t) \, dt, \tag{12}$$

*and assume the Riemannian metric is given by Eq. 5. The length of a path is given by Eq. 6. Suppose that the geodesic path connecting $\overline{\mathbf{x}}$ and $\mathbf{x}$ exists and that the integrand $\frac{\partial f}{\partial x_i}(\gamma(t)) \, \dot{\gamma}_i(t)$ is continuous in $t$ for each $i$. Then,*

$$\sum_{i=1}^n \left| A_i^{\gamma}(\mathbf{x}) \right| = \left| f(\mathbf{x}) - f(\overline{\mathbf{x}}) \right|$$

*if and only if $\gamma$ is the geodesic path.*

See Appendix A for the proof.

### 2.5.2 Symmetry preserving of Geodesic IG

The symmetry axiom is defined in the following way.

**Axiom 3 (Symmetry)** *Consider an input-baseline pair $\boldsymbol{x}$ and $\overline{\boldsymbol{x}}$, and a function $f$ that is symmetric in dimensions $i$ and $j$. If $\boldsymbol{x}_i = \boldsymbol{x}_j$ and $\overline{\boldsymbol{x}}_i = \overline{\boldsymbol{x}}_j$, then an attribution method is Symmetry-Preserving if $A_i(\boldsymbol{x}) = A_j(\boldsymbol{x})$, where $A_n(\boldsymbol{x})$ is the attribution of $\boldsymbol{x}_n$.*

(Sundararajan et al., 2017, Theorem 1) shows that IG is the only path-based attribution method that satisfies symmetry for any function. However, as noted in Kapishnikov et al. (2021), while the straight path is the only one satisfying symmetry for *any* function, for a specific function, it may be possible to find other paths that also satisfy symmetry. Below, we demonstrate that Geodesic IG satisfies symmetry for Riemannian manifolds, and thus for the neural network functions we use when sampling the paths.

Let the $i$th and $j$th dimensions of $\gamma(t)$ be $\gamma_i(t)$ and $\gamma_j(t)$ respectively and $f$ be a function differentiable almost everywhere on $t$. Furthermore, take $f$ to be symmetric with respect to $x_i$ and $x_j$. If $\gamma_i(t) = \gamma_j(t)$ for all $t \in [0, 1]$, then we have

$$||\partial_t f(\gamma_i(t)) \times \dot{\gamma}_i(t)|| = ||\partial_t f(\gamma_j(t)) \times \dot{\gamma}_j(t)||, \tag{13}$$

almost everywhere on $t$. Therefore, the $i$th and $j$th components of Eq. 6 are equal. Furthermore, since Eq. 9 integrates along the path that is an approximation of Eq. 6, we have Geodesic $IG_i$ = Geodesic $IG_j$. Indeed our geodesic paths satisfy $\gamma_i(t) = \gamma_j(t)$ for all $t \in [0,1]$ on the Riemannian manifolds. To see this, let us select a baseline $\bar{\mathbf{x}}$ and $U$ a Riemann neighbourhood centred at $\bar{\mathbf{x}}$. Let us also define the geodesic path $\gamma$ such as $\gamma(0) = \bar{\mathbf{x}}$. Further, define $\mathbf{v}(t) := \gamma'(t)$, where $\gamma'$ is the derivative of $\gamma$. Then, in the local coordinates system of the neighbourhood of any point, called normal coordinates, we have $\gamma(t) = (tv_1(t), ..., tv_n(t))$. Since the function is symmetric in the $i$th and $j$th dimensions, we have $v_i$ and $v_j$ are the same everywhere. From this, we can see that $\gamma_i(t) = \gamma_j(t)$ for all $t \in [0,1]$ and therefore Geodesic IG satisfies symmetry.

## 3 Experiments

To validate our method, we performed experiments on two datasets: one is the synthetic half-moons dataset, and the other is the real-world Pascal VOC 2012 dataset.

### 3.1 Experiments on the half-moons dataset

We use the half-moons dataset provided by Scikit learn (Pedregosa et al., 2011) to generate 10,000 points with a Gaussian noise of $\mathcal{N}(0, x)$, where $x$ ranges between 0.05 and 0.65. The dataset is split into 8,000 training points and 2,000 testing ones. The model used is an MLP.

We evaluate each attribution method using an indicator of performance: the absence of artefacts that do not reflect the model's behaviour. To this end, we use purity, defined as follows.

A well-trained model should classify approximately half of the data points as "upper moon" (class 1) and the other half as "lower moon" (class 0). Such a model should consider both features of each point important for classification into class 1. Therefore, for a good attribution method, A, we expect the top 50% of points, ranked by the quantity $\widetilde{A}(\mathbf{x}) = \sum_{i=0}^{1} |A_i(\mathbf{x})|$, to be classified as 1, assuming the baseline is chosen as a point to which the network assigns a near-zero score. With this in mind, we define purity as

$$\text{Purity} = \frac{1}{N/2} \sum_{\mathbf{x}, \, \widetilde{A}(\mathbf{x}) \in \text{Top 50\% of all A}} \text{argmax}(f(\mathbf{x})), \tag{14}$$

where $N$ is the number of data points. For any fixed noise level, purity ranges in $[0,1]$, where values near 1 indicate that the points with largest total attribution are mostly classified as class 1, while a random attribution ranking yields purity near 0.5.

**From Purity to AUC-Purity.** Because we evaluate across multiple noise levels, we need a single summary statistic. In Fig. 8 we plot purity as a function of noise level, and in Table 1 we summarise each method by the area under that purity–noise curve (AUC-Purity) over the noise range $[0.05, 0.65]$. Under this summary, a random method yields approximately $0.5 \times (0.65 - 0.05) \approx 0.30$, matching the Random baseline in Table 1. Higher AUC-Purity values indicate better performance across all noise levels.

In this experiment, we compare the results of attributions from Geodesic IG with methods including Integrated Gradients, GradientShap, InputXGradients (Shrikumar et al., 2016), KernelShap (Lundberg and Lee, 2017), Occlusion (Zeiler and Fergus, 2014), and Guided IG (Kapishnikov et al., 2021). We note that KernelShap is the kernel-based approximation of SHAP values (Lundberg and Lee, 2017), and thus our comparison already covers the SHAP family. We do not separately include LIME (Ribeiro et al., 2016) because LIME and KernelShap are closely related (both fit local linear models to perturbed inputs) and KernelShap has been shown to converge to the same Shapley values that LIME approximates (Lundberg and Lee, 2017). Including both would add little additional discriminative information.

For all of the methods, we use $(-0.5, -0.5)$ as a baseline. The chosen number of neighbours for the $k$NN part of both Enhanced IG and Geodesic IG is 15.

We have run our experiment with 5 different seeds and plotted the mean and standard error of the results for different noise levels in Fig. 8. We have summarised these results by reporting their area under the curve (AUC) in Table 1. We see that our $k$NN-based Geodesic IG outperforms all other methods, with the

| Method | AUC-Purity ↑ |
|---|---|
| Input X Gradients | 0.328 |
| GradientShap | 0.483 |
| IG | 0.487 |
| Random | 0.299 |
| Kernel Shap | 0.480 |
| Occlusion | 0.520 |
| Guided IG | 0.361 |
| Enhanced IG | 0.470 |
| Geodesic IG ($k$NN) | **0.531** |
| Geodesic IG (SVI) | 0.504 |

Table 1: Evaluation of different attribution methods on a half-moons dataset with Gaussian noises with standard deviation ranging from 0.05 to 0.65. While our $k$NN-based method outperforms all other methods, we see that unlike larger examples, such as the one summarised in Table 2, our SVI example struggles to compete due to complexity of tuning hyperparameters.

gap increasing with noise. While Occlusion comes close second, we see in Table 2 that the method does not perform well in larger examples with more complex embeddings. To provide better understanding of the comparison of our results with Enhanced IG we present more analysis on this dataset in Section 4.

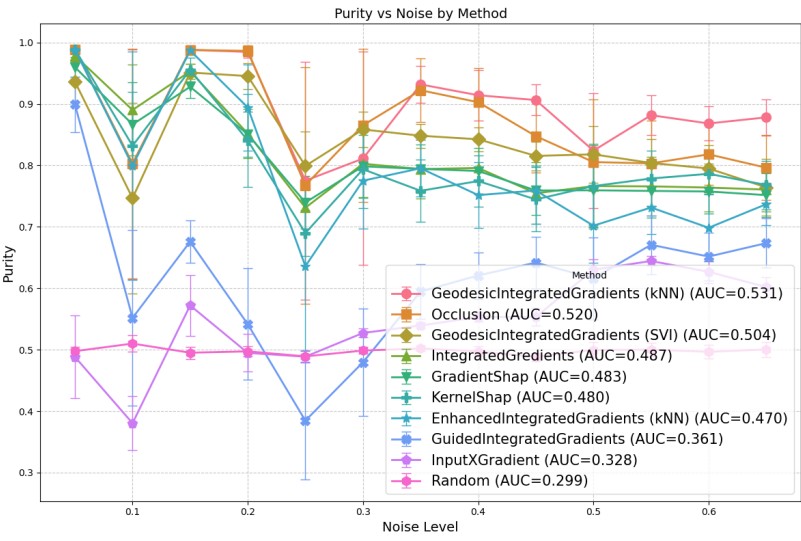

Figure 8: A comparison of different attribution methods on half-moons dataset with noise levels ranging from 0.05 to 0.65. Our $k$NN-based method outperforms all, with the gap increase with noise.

## 3.2 Experiments on the Pascal VOC 2012 dataset

To evaluate our method on a real-world dataset, we used the Pascal VOC 2012 dataset (Everingham et al.), which consists of labelled images. We trained a classification head on this dataset and integrated it with the pre-trained ConvNext model (Liu et al., 2022) from TorchVision to generate predictions for explanation.

For this experiment, we used the same attribution methods as our half-moons experiment, except for the $k$NN-based methods (Enhanced IG and Geodesic IG ($k$NN)) since densely sampling and searching the gradient space of a large model was not practical. We applied these attribution methods to classification of 100

| Method | AUC-Comp ↑ | AOC-LO ↑ | Time (s/image) |
|---|---|---|---|
| Input X Gradients | 0.21 | 1.28 | < 1 |
| GradientShap | 0.18 | 1.15 | ∼ 2 |
| IG | 0.21 | 1.25 | ∼ 1 |
| Random | 0.16 | 0.86 | – |
| Kernel Shap | 0.16 | 0.94 | ∼ 60 |
| Occlusion | 0.19 | 1.16 | ∼ 30 |
| Guided IG | 0.20 | 1.18 | ∼ 5 |
| Geodesic IG (SVI) | **0.27** | **1.44** | ∼ 840 |

Table 2: Evaluation of different attribution methods on 100 randomly sampled images from the Pascal VOC test set. Fig. 2 shows the curves where these metrics are extracted from. Runtime estimates are approximate and measured on an NVIDIA L4 GPU. Results are reported without confidence intervals due to computational constraints; see Section 3.2 for discussion of effect sizes.

randomly selected images. For explainers that require a baseline, such as IG and our proposed method, a uniformly black image was used as the baseline.

We note that the sample size of 100 images is motivated by the substantial computational cost of the SVI-based method (approximately 14 minutes per image on an NVIDIA L4 GPU, totalling 23 GPU-hours for 100 images). While a larger sample would reduce variance in our estimates, we observe that our method's improvement over the next best baseline is large: a relative improvement of approximately 29% in AUC-Comprehensiveness (0.27 vs. 0.21) and 15% in AOC-Log-odds (1.44 vs. 1.28). These effect sizes are substantial enough to be statistically meaningful even at $n = 100$. Furthermore, the consistency of improvement across both metrics, across different masking percentages $k$ (Fig. 2), and across the qualitative examples (Fig. 1 and Appendix B) provides converging evidence that the improvement is robust rather than artefactual. We view extending to larger-scale evaluations as a worthwhile direction once more efficient geodesic solvers are available (see Discussion).

**Scope of evaluation.** Our real-world evaluation focuses on Pascal VOC 2012 with ConvNext to provide a controlled comparison across methods. While the theoretical framework applies to any differentiable model, validating GIG across additional domains (e.g., NLP, tabular data) and architectures (e.g., ResNet, Vision Transformers) is a natural direction for future work as more efficient geodesic solvers reduce the per-image cost. We encourage future work to evaluate GIG across a broader set of benchmarks and to include standard evaluation protocols such as insertion/deletion curves (Shrikumar et al., 2017) and ROAR (Hooker et al., 2019) alongside the metrics we report.

To measure the performance of an attribution method, we use 2 different metrics:

- **Comprehensiveness** (DeYoung et al., 2019): We mask the top k% most important features in absolute value, and compute the average change of the predicted class probability compared with the original image. A higher score is better as it indicates masking these features results in a large change of predictions.

- **Log-odds** (Shrikumar et al., 2017): We mask the top k% most important features in absolute value, and measure the log-odds of the target class. At each masking level, more negative log-odds indicates stronger removal of evidence (better). We summarise this as AOC-Log-odds (see below), where higher is better.

We evaluated these metrics for a range of top-$k$%, from 1% to 65%, as shown in Fig. 2. To summarise performance across different $k$% values, we calculated the area under the curve (AUC) for Comprehensiveness (higher AUC = better, since higher comprehensiveness at each $k$ is better).

For Log-odds, interpretation requires care. At each masking level $k$, the raw log-odds value is typically negative (masking important features reduces evidence for the target class), and more negative values indicate better attribution quality. To produce a summary where "higher is better" for both metrics, we report the area between the log-odds curve and zero, equivalently, the magnitude of the area under the (negative) curve.

We call this the area-over-curve (AOC). Under this convention, a method that produces more negative log-odds after masking will have a larger AOC, so **higher AOC-Log-odds is better** in Table 2, consistent with the AUC-Comprehensiveness column.

Additionally, Fig. 1 provides a qualitative comparison between Geodesic IG and Integrated Gradients. The results demonstrate that Geodesic IG outperforms other methods in explaining the model's behaviour on the dataset, with a particularly notable improvement in comprehensiveness. Further qualitative comparisons can be found in Appendix B.

Using Geodesic IG to explain complex deep learning models comes with practical trade-offs. The energy-based method requires iterative optimisation and therefore incurs substantially higher compute than standard IG-style line integration. In our Pascal VOC experiment, explaining 100 images required approximately 23 GPU-hours on an L4 (about 14 minutes per image). To contextualise this cost, we include approximate per-image runtimes for all methods in Table 2: standard IG requires approximately 1 second, Guided IG approximately 5 seconds, KernelShap approximately 60 seconds, and Geodesic IG (SVI) approximately 840 seconds. The cost of our method is thus roughly $14\times$ that of KernelShap and $840\times$ that of IG. This cost is acceptable in settings where high-quality explanations are required (e.g., debugging, auditing, or safety-critical review), but may be prohibitive for real-time use. Efficiency can be improved by reducing the number of optimisation steps, warm-starting paths from neighbouring images, or solving the geodesic ODE directly rather than via variational inference (see Discussion). We therefore view the current method as a high-fidelity explainer rather than a lightweight saliency tool, and we emphasise that the theoretical contribution (NCC axiom and its characterisation of geodesic paths) is independent of the particular solver used.

Another challenge is that the performance of the energy-based geodesic method depends on selecting the right value for $\beta$ in Eq. 11 and tuning SVI hyperparameters, such as the learning rate. In principle, hyperparameter tuning using metrics like Comprehensiveness or Log-odds could help optimise these parameters. However, due to limited computational resources, we were unable to perform such tuning in this study, though it has the potential to significantly improve results. Lastly, the optimisation nature of these sampling methods can cause the endpoints of the paths to deviate slightly from the baseline and input points, favouring nearby points in lower gradient regions. To address this, we added a term to the potential energy to correct for these deviations and ensure more accurate alignment with the intended points. Formally, the extra term is $w \sum_{t \in \text{endpoints}} |\gamma(t) - \gamma_{\text{init}}(t)|_2$, where $w$ is endpoint weight and the first/ last 10% of the paths are counted as endpoints .

## 4    Related Work

Approximating geodesic paths is a widely studied area of research, and many methods to do so have been developed. For a comprehensive survey on this subject, please refer to Crane et al. (2020).

Recent work has revisited path choice for Integrated Gradients. Denoising Diffusion Path (Lei et al., 2024) targets noise in path-based attribution by using an auxiliary diffusion model to construct a piecewise path whose intermediate points remain near the training distribution, reducing accumulated attribution noise. A complementary direction modifies where along a baseline-to-input path gradients are accumulated. Integrated Decision Gradients (Walker et al., 2024) addresses saturation by emphasising the portion of the path where the model output changes most rapidly ("where the model makes its decision"), improving faithfulness without requiring a learned manifold.

The idea of using a $k$NN algorithm to avoid computing gradients on out of distribution data points has also been used in Enhanced Integrated Gradients Jha et al. (2020). However, this method creates a path which is model agnostic, as it does not necessarily avoid high gradients regions. As a result, it can lead to significant artefacts which do not reflect the model's behaviour. To support this argument, we provide an example where this method fails on the half-moons datasets. In Fig. 9, similar to the example in section 1, we see the Enhanced IG attributes different importance to the horizontal and vertical features of the half-moon data points, in a model that is flat everywhere, other than the decision boundary. Furthermore, in Fig. 9, we observe that the method violates NCC, Axiom 2. This is expected, given Theorem 1.

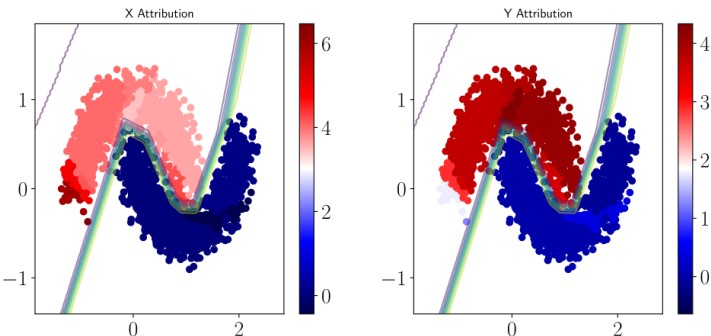

Figure 9: **Enhanced IG attributions.** Enhanced IG applies a *k*-NN algorithm, then uses Dijkstra's algorithm to find the shortest path between an input and a reference baseline, and computes gradients along this path. However, since this method is model-agnostic, it does not account for high-gradient regions. As a result, the attributions exhibit an unjustified shading that does not accurately reflect the model's true behaviour. Comparing the horizontal and vertical attributions (left and right plots, respectively), we confirm that this method does not satisfy Axiom 2, as expected. In this example, similar to those in Fig. 3, we sample 10,000 points from the half-moons dataset with noise drawn from $\mathcal{N}(0, 0.15)$.

Guided Integrated Gradients (Kapishnikov et al., 2021) adapts the path with heuristics intended to reduce noise. Blur Integrated Gradients (Xu et al., 2020) replaces the straight path in pixel space with a path in "scale space," progressively sharpening a blurred version of the input; this provides a different inductive bias (scale rather than geometry) for path construction. In contrast, our method directly approximates geodesics under a model-induced Riemannian metric, targeting the path of least resistance with respect to the explained model's gradient landscape.

**Connection to information geometry.** Our model-induced metric $G_{\mathbf{x}} = J_{\mathbf{x}}^T J_{\mathbf{x}}$ is closely related to the Fisher Information Metric (FIM) used in information geometry (Dombrowski et al., 2019). For probabilistic models, the FIM captures the local curvature of the log-likelihood surface and has been used to study the robustness and geometry of neural network explanations. While the FIM is defined over parameter space and requires probabilistic outputs, our metric operates directly in input space and is defined for any differentiable scalar function. The conceptual link is that both metrics use the Jacobian to endow a flat space with curvature that reflects the function's local sensitivity.

**Perturbation-based methods.** SHAP (Lundberg and Lee, 2017) and LIME (Ribeiro et al., 2016) take a fundamentally different approach to attribution: rather than integrating gradients along a path, they perturb inputs and observe output changes, fitting local surrogate models to estimate feature importance. Our experiments include KernelShap (the kernel-based SHAP approximation) as a representative of this family. While perturbation-based methods are model-agnostic and do not require gradient access, they suffer from known limitations: KernelShap's computational cost scales poorly with the number of features (as seen in Table 2), and the choice of perturbation distribution can significantly affect results (Sundararajan et al., 2017). Our path-based approach is complementary: it requires gradient access but provides attributions grounded in the model's differential geometry. The two paradigms address different aspects of faithfulness, and we include KernelShap in our experiments to enable cross-paradigm comparison.

**Relationship to Manifold Integrated Gradients.** Closest to our work is the Manifold Integrated Gradients (MIG) framework of Zaher et al. (2024), which also integrates model gradients along geodesic paths derived from a Riemannian metric. Both methods share the core insight that Riemannian geometry provides a principled foundation for improving path-based attributions, and MIG demonstrated convincingly that geodesic paths yield perceptually cleaner and more robust feature visualisations. The two methods differ primarily in the choice of Riemannian structure, which leads to complementary strengths. MIG defines its

metric through the Jacobian of a VAE decoder, $G_g = J_g^\top J_g$, so that geodesics respect the curved geometry of the data manifold learned by the generative model. Our method instead equips the input space with the metric $G_{\mathbf{x}} = J_{\mathbf{x}}^\top J_{\mathbf{x}}$ induced by the Jacobian of the explained model itself, so that geodesics follow the path of least resistance through the model's gradient landscape (Remark 1).

This distinction leads to different practical trade-offs. MIG's data-geometric perspective offers the appealing property that geodesic interpolants remain on the data manifold, producing realistic intermediate images and providing inherent robustness to adversarial attributional attacks, as demonstrated in their experiments. Because MIG's metric is derived from a VAE decoder rather than the classifier, the geodesic path captures data-distributional structure and is shared across any model trained on the same data. Our model-geometric perspective instead tailors the path to the specific classifier being explained, directly targeting the high-gradient regions that cause attribution artefacts. A practical difference is that MIG requires training an auxiliary VAE and retraining the classifier on VAE reconstructions, while our approach requires no generative model and explains the original classifier directly. Our work also contributes a formal characterisation that complements MIG's empirical findings: we introduce the NCC axiom (Axiom 2) and prove (Theorem 1) that, under the model-induced metric, NCC holds if and only if the integration path is a geodesic.

Because the two methods use different experimental setups (MIG uses Oxford Pets and Oxford Flowers with a VAE-based pipeline and evaluates robustness via SSIM under adversarial attack, while we use Pascal VOC with comprehensiveness and log-odds metrics), a controlled head-to-head comparison would require reproducing both pipelines in full. We therefore focus on clarifying the conceptual relationship rather than attempting a potentially misleading empirical comparison. In summary, MIG and our method represent complementary perspectives on how Riemannian geometry can improve path-based attributions: MIG leverages the geometry of the data to keep interpolants realistic, while our method leverages the geometry of the model to keep attributions faithful. We believe the two ideas could fruitfully be combined, for instance by defining a metric that incorporates both data-manifold and model-gradient information, and we view this as a promising direction for future work.

## 5 Discussion

In this paper, we identified key limitations of path-based attribution methods such as Integrated Gradients (IG), particularly the artefacts that arise from ignoring the model's curvature. To address these issues, we introduced a novel path-based method, Geodesic IG, that integrates gradients along geodesic paths on a manifold defined by the model, rather than straight lines. Our work is complementary to the Manifold Integrated Gradients (MIG) framework of Zaher et al. (2024), which demonstrated the value of geodesic paths from a data-manifold perspective. While MIG derives its Riemannian metric from a VAE decoder to keep interpolants on the data manifold, our model-induced metric targets high-gradient regions of the explained model directly and requires no auxiliary generative model. The two perspectives address different aspects of attribution quality and could be combined in future work.

We emphasise that the core contribution of this work is conceptual and theoretical: the identification of a model-induced Riemannian structure that characterises when and why geodesic paths are uniquely suitable for attribution (Theorem 1), rather than any particular geodesic solver. The kNN and SVI methods we present are proof-of-concept implementations; as more efficient solvers become available, the theoretical framework will apply unchanged.

By avoiding regions of high gradient in the input space, Geodesic IG effectively mitigates these artefacts while preserving all the axioms established by Sundararajan et al. (2017). Additionally, we introduced a new axiom, No-Cancellation Completeness (NCC), which, when satisfied, prevents such misattributions. We proved that, for path-based attributions in Eq. 12 under the model-induced metric in Eq. 5, NCC holds if and only if the integration path is a geodesic (Theorem 1), subject to mild regularity conditions. Through both theoretical analysis and empirical evaluation on synthetic and image classification benchmarks, using metrics such as Comprehensiveness and Log-Odds, we provided evidence of the advantages of our approach.

To approximate geodesic paths, we proposed two methods: one based on $k$-Nearest Neighbour and another leveraging Stochastic Variational Inference. While these methods outperform existing alternatives, they

also present challenges. One such challenge is computational cost, as discussed in Section 3. Another is the inherent noise in sampling-based geodesic approximations. Even though in our experiments we demonstrated noise reduction relative to the original IG, we believe further improvements can be achieved. Several promising directions exist for improving scalability:

1. **Direct geodesic ODE solvers:** Rather than using variational inference, one could directly solve the geodesic differential equation $\nabla_{\dot{\gamma}}\dot{\gamma} = 0$ using standard ODE solvers (e.g., Runge-Kutta methods). This avoids the stochastic optimisation loop entirely and could reduce computation by orders of magnitude.

2. **Amortised path prediction:** A neural network could be trained to predict geodesic paths given input-baseline pairs, amortising the per-image optimisation cost into a one-time training cost.

3. **Warm-starting:** For batches of similar images, paths optimised for one image can initialise the optimisation for nearby images, reducing the number of SVI iterations needed.

4. **Reduced-resolution paths:** For image inputs, the geodesic path could be optimised at reduced spatial resolution and then upsampled, since the gradient landscape varies smoothly at coarse scales.

We view developing efficient geodesic solvers as an important direction for making GIG practical at scale.

## 5.1 Limitations

We acknowledge several limitations of our work. First, our experiments focus on image classification with a single real-world dataset (Pascal VOC 2012) and a single architecture (ConvNext); while the theoretical framework applies broadly, we have not empirically validated GIG on other domains such as natural language processing or tabular data, nor on other vision architectures such as ResNets or Vision Transformers. Second, the computational cost of the SVI-based method is substantial (approximately $840\times$ that of standard IG), limiting its applicability to settings where explanation quality is prioritised over speed. Third, while we provide intuition for the approximation quality of our methods, formal approximation bounds relating our discrete paths to true geodesics remain an open theoretical question; in particular, the SVI-based method minimises a heuristic energy function (Eq. 11) rather than the Riemannian length functional directly. Fourth, hyperparameter selection for the SVI method ($\beta$, learning rate) requires tuning, and our reported results may underestimate the method's potential due to limited computational resources for this tuning. Finally, the metric tensor $G_{\mathbf{x}}$ is rank-1 degenerate for scalar-valued models, meaning geodesics are not uniquely defined in flat regions; while this does not cause practical issues in our experiments (see Remark 2), a formal treatment of geodesics under degenerate metrics would strengthen the theoretical foundations.

## 5.2 Attribution under model bias

A natural question is whether attribution methods, including ours, remain meaningful when the model itself is biased, for instance when a classifier relies on spurious correlations or shortcuts. In such cases, different training runs may produce models that focus on different (possibly unrelated) features, and attributions will faithfully reflect whichever features the specific model uses. This is by design: attribution methods aim to explain *what the model does*, not *what the model should do*. Geodesic IG is no exception; it explains the model as-is, and if the model is biased, the attributions will reflect that bias. Indeed, this can be a strength, since faithful attributions of a biased model can help practitioners *diagnose* the bias. Methods that impose data-distribution priors on the path (e.g., staying on the data manifold) offer different trade-offs: they may produce more perceptually natural interpolants, but they may also smooth over model-specific biases by constraining attributions to data-plausible directions. By contrast, our model-induced metric ensures that the attribution path is shaped by the model's own gradient landscape, which can make biases more visible.

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

# A Proof of Theorem 1

**Proof 1** *We proceed in two parts: first, we show that if $\gamma$ is the geodesic path, then the equality holds; second, we show that if the equality holds, then $\gamma$ must be the geodesic path.*

### Sufficiency: If $\gamma$ is the geodesic, then the equality holds.

*Let $\gamma$ be the geodesic path connecting $\overline{\mathbf{x}}$ and $\mathbf{x}$. By definition of a geodesic in the given Riemannian metric, $\gamma$ satisfies the geodesic equation:*

$$\nabla_{\dot{\gamma}}\dot{\gamma} = 0,$$

*where $\nabla$ is the Levi-Civita connection associated with the metric $\mathbf{G_x}$. Under the given metric $\mathbf{G_x} = \nabla f(\mathbf{x})\nabla f(\mathbf{x})^T$, the geodesic equation implies that $\dot{\gamma}(t)$ is parallel to $\nabla f(\gamma(t))$ for all $t \in [0, 1]$. That is,*

$$\dot{\gamma}(t) = \lambda(t)\nabla f(\gamma(t)),$$

*for some scalar function $\lambda(t)$.*

*Substituting this into the definition of $A_i^\gamma(\mathbf{x})$, Eq. 12, we obtain:*

$$A_i^\gamma(\mathbf{x}) = \int_0^1 \frac{\partial f}{\partial x_i}(\gamma(t))\dot{\gamma}_i(t)\, dt = \int_0^1 \frac{\partial f}{\partial x_i}(\gamma(t))\lambda(t)\frac{\partial f}{\partial x_i}(\gamma(t))\, dt.$$

*Since $\lambda(t)$ does not change sign (as $\gamma$ is a geodesic[2]), the integrand $\frac{\partial f}{\partial x_i}(\gamma(t))\dot{\gamma}_i(t)$ maintains a consistent sign across all $i$ and $t$. Therefore,*

$$\sum_{i=1}^n |A_i^\gamma(\mathbf{x})| = \left|\sum_{i=1}^n A_i^\gamma(\mathbf{x})\right|.$$

*By the Fundamental Theorem of Calculus,*

$$\sum_{i=1}^n A_i^\gamma(\mathbf{x}) = \int_0^1 \frac{d}{dt}f(\gamma(t))\, dt = f(\mathbf{x}) - f(\overline{\mathbf{x}}).$$

*Thus,*

$$\sum_{i=1}^n |A_i^\gamma(\mathbf{x})| = |f(\mathbf{x}) - f(\overline{\mathbf{x}})|.$$

### Necessity: If the equality holds, then $\gamma$ is the geodesic.

*Suppose $\gamma$ is a smooth path such that*

$$\sum_{i=1}^n |A_i^\gamma(\mathbf{x})| = |f(\mathbf{x}) - f(\overline{\mathbf{x}})|.$$

*By the triangle inequality,*

$$\sum_{i=1}^n |A_i^\gamma(\mathbf{x})| \geq \left|\sum_{i=1}^n A_i^\gamma(\mathbf{x})\right| = |f(\mathbf{x}) - f(\overline{\mathbf{x}})|.$$

---

[2]Since $\gamma$ is a geodesic, its velocity is always parallel to the gradient of $f$, so there exists a scalar function $\lambda(t)$ such that

$$\dot{\gamma}(t) = \lambda(t)\nabla f(\gamma(t)).$$

Then, by the chain rule,

$$\frac{d}{dt}f(\gamma(t)) = \nabla f(\gamma(t)) \cdot \dot{\gamma}(t) = \lambda(t)\|\nabla f(\gamma(t))\|^2.$$

Assuming $\|\nabla f(\gamma(t))\|^2 > 0$, the sign of $\frac{d}{dt}f(\gamma(t))$ is completely determined by the sign of $\lambda(t)$. Since $f(\gamma(t))$ changes monotonically (i.e., without backtracking) from $f(\gamma(0))$ to $f(\gamma(1))$, $\lambda(t)$ must maintain a constant sign along $\gamma$.

Intuitively, $\dot{\gamma}(t) = \lambda(t)\nabla f(\gamma(t))$ means that $\gamma$ is always moving in the direction of steepest ascent (or descent) of $f$, with $\lambda(t)$ scaling its speed. Any reversal in the sign of $\lambda(t)$ would indicate a change in the direction of motion relative to $\nabla f$, corresponding to backtracking, a behaviour that contradicts the optimality of a geodesic. Thus, the monotonic variation of $f(\gamma(t))$ ensures that $\lambda(t)$ does not change sign.

Equality holds if and only if all $A_i^\gamma(\mathbf{x})$ share the same sign. This implies that for each $i$, the integral $\int_0^1 \frac{\partial f}{\partial x_i}(\gamma(t))\dot{\gamma}_i(t)\,dt$ has a definite sign. Since the integrand $\frac{\partial f}{\partial x_i}(\gamma(t))\dot{\gamma}_i(t)$ is continuous in $t$ (by our regularity assumption), a definite sign of the integral implies that the integrand itself does not change sign[3]. Consequently, $\nabla f(\gamma(t)) \cdot \dot{\gamma}(t)$ does not change sign, and $\dot{\gamma}(t)$ is everywhere parallel to $\nabla f(\gamma(t))$. That is,

$$\dot{\gamma}(t) = \lambda(t)\nabla f(\gamma(t)),$$

for some scalar function $\lambda(t)$.

Under the given Riemannian metric $\mathbf{G_x} = \nabla f(\mathbf{x})\nabla f(\mathbf{x})^T$, such paths are precisely the geodesics, connecting $\overline{\mathbf{x}}$ and $\mathbf{x}$.

**Conclusion:** Combining the Necessity and Sufficiency parts above, we can conclude that the equality $\sum_{i=1}^n |A_i^\gamma(\mathbf{x})| = |f(\mathbf{x}) - f(\overline{\mathbf{x}})|$ holds if and only if $\gamma$ is the geodesic path connecting $\overline{\mathbf{x}}$ and $\mathbf{x}$.

---

[3]If a continuous function $g(t)$ satisfies $|\int_0^1 g(t)\,dt| = \int_0^1 |g(t)|\,dt$, then $g$ does not change sign. Indeed, if $g$ changed sign, continuity would imply a set of positive measure where $g$ has opposite sign, making the inequality strict.

# B  Additional heatmaps and results on Pascal VOC 2012

We also qualitatively compare on Figure 10 Geodesic IG with the original IG on 5 different images of the Pascal VOC 2012 dataset. In these images Geodesic IG heatmaps appears to have fewer artefacts and is not sensitive to the choice of baseline being a black image. This is contrary to IG, which assigns no importance to the segments of the image that are black, since they have no difference to the chosen baseline.

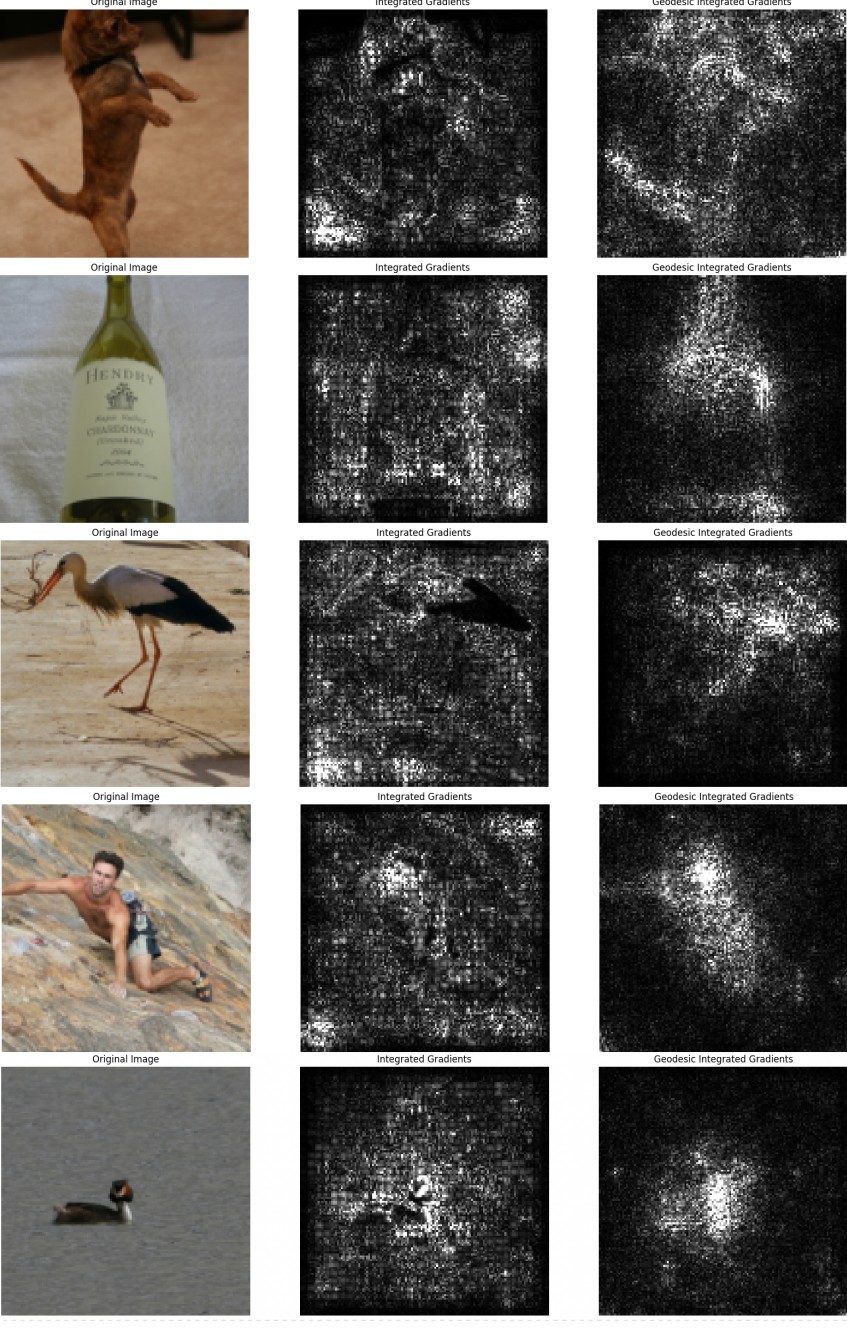

Figure 10: Heatmaps of Integrated Gradients (middle) and Geodesic IG (right) on 5 images from the test set of Pascal VOC 2012.

