# OpenReview forum: "Using the Path of Least Resistance to Explain Deep Networks"
_TMLR — Withdrawn by Authors_

### Review · Reviewer_FEye · 2026-03-24

**Summary Of Contributions:**

The paper investigates limitations of Integrated Gradients (IG), an axiomatic path-based attribution method that assigns feature importance by integrating gradients along a straight-line path from a baseline to the input. To address these limitations, the authors propose Geodesic Integrated Gradients (GIG), a variant of IG that equips the input space with a model-induced Riemannian metric and computes attributions along geodesic paths. These paths are approximated using kNN-based methods for smaller models and stochastic variational inference for larger ones. Additionally, the paper introduces a new axiom, No-Cancellation Completeness (NCC), which strengthens the standard completeness property by preventing feature-wise cancellation effects.

**Audience:**

Yes

**Audience Explanation:**

The topic treated and the idea proposed are interesting.

**Claims And Evidence:**

No

**Claims Explanation:**

The experimental section focuses primarily on quantitative evaluation metrics, with limited qualitative analysis (aside from Figure 1). Given that the goal of the method is to improve explanations, the absence of qualitative examples and to only mention  the importance of a human-centered evaluation is a notable limitation. Understanding how users perceive and benefit from these explanations is essential in the XAI context. Furthermore, the experiments are conducted on only two datasets. This is insufficiently justified, especially considering that the proposed method is general-purpose and not tailored to a specific domain. A broader evaluation across diverse datasets would be necessary to support claims of general applicability. Another important limitation is the absence of statistical significance analysis. The reported improvements are not supported by tests such as critical difference diagrams with Nemenyi post-hoc analysis (or similar), which weakens the reliability of the empirical conclusions. While the inclusion of a large number of competitors is appropriate, the presentation of results raises concerns. In particular, Figure 8 is difficult to interpret due to overlapping legends, suboptimal color choices, and line thickness. The lack of visual clarity makes it challenging to verify the claimed performance improvements. Combined with the limited number of datasets and absence of statistical testing, it is not possible to confidently conclude that the proposed method consistently outperforms existing approaches. The use of multiple evaluation metrics is appropriate and commendable; however, relying on a single dataset per type for some of these evaluations undermines their validity.

**Requested Changes:**

The abstract clearly communicates the contributions and motivations of the work. The experimental results suggest that GIG is more faithful than IG, making it crucial to carefully examine how faithfulness is defined and evaluated.

The motivating examples presented in the introduction are interesting and help contextualize the problem. However, the overall organization of the introduction significantly hinders readability:

- The introduction is excessively long and could be streamlined.
- There is a notable misalignment between textual references and figure placement. For instance, Figure 2 is referenced after Figure 3, yet appears earlier, creating confusion.
- Page 4 is particularly difficult to follow, as it combines large figures, extended captions, and the introduction of Axiom 1 with a central formula, resulting in a visually and cognitively dense layout.

A restructuring is strongly recommended. Parts of the current introduction could be moved to later sections (e.g., methodology or into a stand-alone section) as motivating examples, allowing the introduction to focus on high-level framing. While subjective, placing the related work after the introduction would also improve the logical flow.

Additionally, it is somewhat unusual to include a paragraph titled “Evaluation Metrics”- within the Method section, as this content typically belongs to the experimental setup.

The methodological presentation is generally clear and appears theoretically sound. However, the proposal places significant emphasis on distance functions and path construction mechanisms that are, in themselves, not novel. At a high level, the method can be interpreted as an extension of IG where the straight-line interpolation between baseline and input is replaced by alternative, geometry-aware paths. While this is a meaningful conceptual contribution, the degree of novelty from an algorithmic perspective may be perceived as incremental.

The experimental section focuses primarily on quantitative evaluation metrics, with limited qualitative analysis (aside from Figure 1). Given that the goal of the method is to improve explanations, the absence of qualitative examples and to only mention  the importance of a human-centered evaluation is a notable limitation. Understanding how users perceive and benefit from these explanations is essential in the XAI context. Furthermore, the experiments are conducted on only two datasets. This is insufficiently justified, especially considering that the proposed method is general-purpose and not tailored to a specific domain. A broader evaluation across diverse datasets would be necessary to support claims of general applicability. Another important limitation is the absence of statistical significance analysis. The reported improvements are not supported by tests such as critical difference diagrams with Nemenyi post-hoc analysis (or similar), which weakens the reliability of the empirical conclusions. While the inclusion of a large number of competitors is appropriate, the presentation of results raises concerns. In particular, Figure 8 is difficult to interpret due to overlapping legends, suboptimal color choices, and line thickness. The lack of visual clarity makes it challenging to verify the claimed performance improvements. Combined with the limited number of datasets and absence of statistical testing, it is not possible to confidently conclude that the proposed method consistently outperforms existing approaches. The use of multiple evaluation metrics is appropriate and commendable; however, relying on a single dataset per type for some of these evaluations undermines their validity.

In the discussion, the authors state that the core contribution of the work is conceptual and theoretical. However, this positioning is not clearly communicated from the outset. Moreover, while the theoretical contribution is potentially valuable, there appears to be no fundamental obstacle preventing a more extensive empirical validation. Although the limitations section acknowledges some of these aspects, they are not sufficiently justified. Presenting a method with strong practical implications while limiting its evaluation to a narrow empirical scope raises concerns about its readiness for publication in its current form. If the primary goal is to introduce a theoretical framework, this should be explicitly stated early in the paper and aligned with the expectations of the target venue.

The paper presents an interesting and potentially impactful idea, supported by a solid theoretical foundation. However, significant issues remain in terms of clarity, organization, and empirical validation. The limited experimental scope, absence of statistical significance analysis, and lack of qualitative or human-centered evaluation prevent a thorough assessment of the method’s practical usefulness. In its current form, the paper does not provide sufficient evidence to support its claims and would benefit from substantial revisions.

---

### Review · Reviewer_jKvs · 2026-03-29

**Summary Of Contributions:**

This paper studies a central issue in path-based attribution methods, namely that standard Integrated Gradients (IG) integrates along a Euclidean straight line from baseline to input, which may traverse regions of high model sensitivity and thereby produce misleading feature attributions. To address this, the authors propose Geodesic Integrated Gradients (GIG), which equips the input space with a model-induced metric and seeks to integrate gradients along geodesic paths under this metric rather than along straight Euclidean interpolations. The paper introduces two approximation strategies: a kNN shortest-path construction for low-dimensional settings and an energy-based SVI procedure for high-dimensional inputs such as images. In addition, the paper proposes a new axiom, No-Cancellation Completeness (NCC), intended to rule out cancellation across feature attributions, and claims that under the proposed model-induced metric this axiom characterises geodesic paths. Empirically, the authors evaluate the method on a synthetic half-moons task and on Pascal VOC with a ConvNext classifier, where they report improvements over several attribution baselines.

**Audience:**

Yes

**Audience Explanation:**

I believe this paper would be of interest to researchers working on explainability, feature attribution, interpretability, and information geometry. The question the paper asks is relevant: whether the choice of attribution path in IG can be made more faithful to the explained model, and whether paths that avoid high-gradient regions can improve attributions. This is a natural and worthwhile question, and the paper offers both a geometric perspective and an empirical attempt at addressing it.

**Claims And Evidence:**

No

**Claims Explanation:**

* Strengths:

The paper addresses a real and interesting problem, and there is clearly something here that could become a useful contribution. The central concern that straight-line Integrated Gradients can pass through regions of high model sensitivity and thereby yield misleading attributions is well motivated, and the toy examples communicate this intuition effectively.

* Weaknesses:

In its current form, I do not think the paper provides sufficiently accurate, convincing, and clear evidence for the strength of the claims it makes.

1. My main issue is that the paper’s framing is stronger than what is actually established. The method is repeatedly described in terms of Riemannian geometry and geodesics under the metric $G_x = J_x^\top J_x$, but the paper also explicitly acknowledges that this metric is degenerate for scalar-valued outputs. That does not automatically invalidate the overall idea, but it does make the geometric interpretation substantially more delicate than the paper currently suggests. In the present version, the paper leans heavily on this language without fully dealing with its consequences.

2. The gap becomes even more apparent in the practical high-dimensional method. The SVI-based method is not actually computing geodesics under the length functional introduced earlier; instead, it minimises a surrogate energy that trades off closeness to the straight path against a penalty based on $||\nabla f(x)||^2$. The paper itself states that this is only a heuristic approximation and is not equivalent to minimising the Riemannian length functional. On top of that, this gradient-norm term is referred to as a “curvature” term, which is mathematically inaccurate. So in practice, the method is much better described as searching for a low-gradient or least-resistance path than as computing geodesics in a strict sense.

3. I also do not find the kNN-based approximation fully convincing as currently presented. To be fair, the method is not purely Euclidean: the edge weights do depend on model gradients. But the neighbourhood graph itself is still built using Euclidean kNN, so the candidate path family is already constrained by an ambient Euclidean notion of locality before any model-aware geometry is introduced. This makes the approximation only partially model-aware, and the paper does not really discuss the implications of that choice in enough depth.

4. The empirical evidence is also too limited for the scope of the claims. The real-world evaluation is restricted to one architecture, one dataset, and 100 sampled images. The baseline choice (black image) is also narrow, despite baseline sensitivity being central to the motivation of the paper. More importantly, the paper’s core story is about better paths, but in the image setting it never actually shows the paths/interpolants produced by the SVI method. We see attribution maps, but not the high-dimensional paths that are supposed to explain why those maps improve. Relatedly, on the half-moons experiment, the paper reports that the kNN-based version outperforms the SVI-based version. This is concerning, because the kNN construction is only plausible in the low-dimensional setting and would not be expected to remain faithful in high-dimensional input spaces for the reasons discussed above. If the method that performs better is the one tied to the low-dimensional setting, while the method that is actually used in the high-dimensional image experiments is empirically weaker even on the toy problem, that raises further questions about how reliable the SVI approximation is in the regime that matters most.

5. The paper should also discuss the interaction between its low-gradient path criterion and saturation effects. Gradient-based attribution is known to become uninformative in saturated regions, where local gradients may vanish even though the underlying features remain important for the prediction. Since the proposed method explicitly prefers paths through low-gradient regions, it is not obvious that this always improves faithfulness rather than steering the path toward regions where gradient information is weak or misleading.

The current evidence does not yet fully support the paper’s present framing and level of claim. I would definitely be willing to reconsider this assessment if the authors sharpen the positioning/claims, use of Riemannian terminology, and strengthen the empirical validation.

**Requested Changes:**

1. Clarify and substantially qualify the geometric framing.
The paper should more carefully explain the consequences of using the degenerate metric $G_x = J_x^\top J_x$ for scalar-valued outputs. In particular, the current use of standard Riemannian and geodesic terminology is too strong without a more careful treatment. At minimum, the paper should explicitly delimit what is meant formally versus heuristically, and avoid overstating standard Riemannian conclusions in the degenerate setting.

2. Reposition the SVI method more accurately.
The paper should clearly state throughout that the SVI method is not solving the geodesic problem exactly, but rather seeks a low-gradient / least-resistance path through a surrogate energy. The terminology “curvature term” for $\||\nabla f(x)\||^2$ should be corrected. The distinction between the exact theoretical object and the practical solver should be made explicit and consistent across the paper.

3. Clarify the status of the NCC theorem relative to the implemented method.
The paper currently presents the theoretical result around NCC and geodesicity in a way that reads more broadly than what the practical method supports. The theorem is stated for the idealised model-induced metric setting, while the implemented high-dimensional method uses a heuristic energy rather than the length functional directly. The paper should be much clearer about this scope mismatch and either narrow the claims or provide stronger justification connecting theory to practice.

4. Strengthen the experimental validation.
The empirical section should be expanded beyond a single real-world architecture/dataset pair. At minimum, I would like to see additional evaluation across more than one vision architecture and multiple baseline choices for IG-like methods, since baseline sensitivity is central to the paper’s motivating examples and known to be important for path-based attribution.

5. Show the actual image-space paths/interpolants used by the SVI method.
Since the paper’s main claim is that improved attribution comes from improved integration paths, the paper should visualise representative sampled/interpolated image paths for the image setting, and compare them against straight-line IG paths and, if possible, other path-based baselines. Without this, a crucial part of the method’s behaviour remains hidden.

6. Add robustness/sensitivity experiments.
The method depends on gradients, optimisation, and hyperparameters such as $\beta$ and the learning rate. The current paper acknowledges the lack of formal sensitivity analysis. I believe some empirical robustness checks are necessary, including sensitivity to baseline choice, hyperparameter sensitivity, and sensitivity to small input perturbations. These are important for assessing whether the method is stable and generalisable.

7. Better discuss the kNN approximation.
The paper should more clearly explain that the kNN graph uses Euclidean neighbourhoods while the edge weights are model-aware, and discuss the implications of this hybrid construction. This would improve the conceptual positioning of the low-dimensional approximation.

Please refer to the weaknesses above, which contain more additional concerns/questions to address during the rebuttal

Minor changes:

- Figures could be improved overall. Some figures are previewed before the relevant concepts and metrics are fully explained, which makes the reading flow less smooth than it could be.
- Several captions should be more self-contained so that the figures can be interpreted without searching through the main text.
- Figure 7 would benefit from clearer annotation and more explicit explanation of its colour coding / scales.
- Figures 2 and 8 are too crowded; the in-plot legend causes occlusion, and separating metrics (for figure 8) into different plots would improve readability.

---

### Review · Reviewer_HFwh · 2026-04-13

**Summary Of Contributions:**

The paper proposes Geodesic Integrated Gradients (GIG), a path-based attribution method that replaces the straight path in Integrated Gradients (IG) with a geodesic path under a model-induced Riemannian metric. The authors introduce a new axiom, No-Cancellation Completeness (NCC), and prove that under their metric, NCC holds iff the path is geodesic. Two approximation methods are presented: a kNN-based approach for low-dimensional inputs and an SVI-based energy minimization for high-dimensional inputs. Experiments on synthetic half-moons and Pascal VOC 2012 with ConvNext are reported, claiming improved faithfulness over IG and several baselines.

**Key Strengths**

+ The theoretical motivation (avoiding high-gradient regions) is sound, and the NCC axiom is a thoughtful extension of completeness.

+ The connection to Riemannian geometry is mathematically elegant.

+ The half-moons example clearly illustrates a failure mode of straight-line paths.

**Key Weaknesses**

- Missing critical baselines. The paper compares against IG, Guided IG, KernelSHAP, etc., but ignores IDGI (Yang et al., CVPR 2023) – a direct variant of IG specifically designed to remove explanation noise via an adaptive path. IDGI is highly relevant and should be compared or at least discussed. Omitting it weakens the empirical claim of superiority.

- Limited evaluation. Only one real-world dataset (Pascal VOC 2012) and one architecture (ConvNext) are used. No results on: Other vision models (ResNet, ViT) or interpretability metrics, e.g., ImageNet with insertion/deletion, ROAR); The paper’s own SVI method underperforms on half-moons (Table 1) and is only tested on 100 images with high variance potential.

- Computational cost not adequately justified. GIG (SVI) takes ~840 seconds per image vs. ~1 sec for IG. The paper acknowledges this but does not show a correspondingly large improvement in faithfulness that would justify such cost for most practical settings. The reported gains (e.g., AUC comprehensiveness 0.27 vs 0.21) are modest relative to the 840x slowdown.

- Theoretical gap in SVI approximation. The energy function (Eq. 11) is a heuristic, not derived from the geodesic length functional (Eq. 6). Theorem 1 applies only to true geodesics, but the SVI paths are not guaranteed to be geodesics. No error bounds or convergence analysis is provided.

- Metric degeneracy glossed over. The rank-1 metric is degenerate in flat regions, making geodesics non-unique. The paper adds a heuristic regularization but does not analyze its effect on attributions or the NCC theorem.

- Limited ablation / hyperparameter analysis. No sensitivity analysis for k or beta  in SVI, or number of interpolation steps. The SVI results are reported without confidence intervals, and tuning was not performed due to “limited computational resources.”

**Audience:**

Yes

**Audience Explanation:**

The paper addresses a core problem in interpretability (path-based attribution) and proposes a geometric solution. This topic is relevant to TMLR’s audience. However, the execution is insufficient for publication: the evaluation is narrow, a key related method (IDGI) is omitted, the computational cost is prohibitive without commensurate gains, and the SVI approximation lacks theoretical guarantees.

**Broader Impact Concerns:**

None significant. The method aims to improve interpretability, which is generally positive. The high computational cost could exacerbate energy consumption for large-scale explanation tasks, but this is not a unique concern. No obvious negative societal impacts.

**Claims And Evidence:**

No

**Claims Explanation:**

1. Incomplete empirical validation

2. Weak evidence for the SVI method

3. Theoretical claim vs. practical implementation gap

4. Incomplete evaluation metrics

**Requested Changes:**

1. Add comparison against IDGI (CVPR 2023) and Manifold Integrated Gradients (Zaher et al., ICML 2024)

2. Include at least two additional models / real-world dataset and report standard metrics (insertion/deletion, ROAR).

3. Provide error bounds or convergence analysis for the SVI approximation relative to true geodesics, or clearly state it as a heuristic and ablate its sensitivity.

4. Ablation study on hyperparameters (e.g., k, beta, number of SVI steps).

---

### Review · Reviewer_QTUq · 2026-04-15

**Summary Of Contributions:**

The authors propose an extension of IG informed by Riemannian geometry, geodesic integrated gradients (GIG). The authors introduce a novel axiom NCC and additional axioms to IGs original set of axioms. The authors prove a base-line attribution method satisfies NCC iff the path is a geodesic. The authors demonstrate the failure of IG on the non-Euclidean dataset half-moon. Two approximation techniques are proposed to select the geodesic path from the induced Riemannian metric; KNN and energy-based. The authors demonstrate GIG outperforms other gradient-based attribution methods under the faithfulness metric on the Pascal dataset. The authors further provide example attributions on the Pascal dataset. The experiments demonstrate GIG better highlights the region of interest in the image classifications.

Strengths of the work:

1)The work and contributions are clearly outlined.

3) The paper is clearly written and easy to follow.

2) The additional axiom is justified and proved. This axiom opens a new avenue for the reliability of other gradient based methods. It further highlights the importance of considering the geometry of the dataset the attributions lie in.

Limitations of the work:

1) Significant overlap with related techniques
As noted by other reviewers there is a significant overlap with GIG and MIG by Zaher et al. both techniques suggest replacing the straight line in IG with a geodesic. MIG is demonstrated to be the robustness to adversarial learning. GIG would need to demonstrate that it outperforms MIG in the adversarial setting.

The computation of the geodesic with KNN is similar to path generation used in discretized integrated gradients (DIG) by Sanyal and Ren. Whilst DIG is in the word embedding domain and GIG is in the Riemannian manifold setting some comparison needs to be made as one may consider word embeddings as lying within a manifold generated by an autoencoder.

Geometrically Guided Integrated Gradients by Rahman et al. propose a similar technique by replacing the path in IG with a path that follows the geometry of the data manifold. The authors should consider this technique and demonstrate where GGIG and the proposed GIG differ.

Costanza and Simpson propose Riemannian Integrated Gradients (RIG). RIG completely generalises path attribution methods to Riemannian manifolds. The authors further abstract each axiom of IG. It may be worth considering if GIG is a special case of RIG and how the propose NCC axiom fits within the Riemannian framework of RIG.

Tangentially Aligned Integrated Gradients (TA-IG) for User-friendly Explanations by Simpson et al. also explore IG in the Riemannian setting. TA-IG selects the base-point of IG such that the explanations lie within the tangent space and demonstrate the explanations provide better attributions than smoothgrad, input*gradient, and IG under common base-point choices. The Simpson et al. demonstrate qualitatively and quantitively with the tangentially alignment metric that the explanations are more user-friendly.

GIG makes a similar claim of better explanations than IG with examples on the Pascal dataset. This claim however is only demonstrated with examples not backed up with qualitative results such as TA-IG.

2) Limited dataset evaluation

Only two datasets are considered. The half-moon dataset demonstrates the pitfalls on IG on non-Euclidean data however a wide array of manifold based datasets should be considered to demonstrate the claims made. The authors have not demonstrated if their results are an artifact of the two datasets or a general result.

The authors evaluated the faithfulness of a variety of gradient-based XAI techniques, but do not demonstrate if the explanations of those techniques are better than GIG.

3) Limited metric evaluation.

Only faithfulness is considered. The authors should consider other metrics such as robustness as proposed by khan et al. or adversarial robustness.

On the earlier point of TA-IG. The tangential alignment metric has been demonstrated by Brodt et al. to be statistically correlated with users reporting better explanations and perceptual alignment. Rather than providing a sample of comparisons between GIG and IG the authors should provide a numerical analysis of the tangential alignment of the explanations.

4) Base-point choices and IG and metrics
The choice of base-point of IG is a known issue in the quality of explanations. Likewise faithfulness and other metrics require a choice of threshold which influences the metrics. A study is need of both the base-points of IG and threshold choices of metrics to provide reliable results.

5) The impact of the geometry. In The Riemannian Geometry of Deep Generative Models Shao et al. demonstrate that the curvature of the data manifolds learned by variational autoencoders (VAEs) for image data are almost flat. It would be interesting the consider the impact of curvature in the datasets considered and how the manifold structure is generated.

If the findings are that the curvature is almost flat this raises questions about the need for the Riemannian approach. Specifically in the energy-based path selection as the curvature term would go to zero.

**Audience:**

No

**Audience Explanation:**

The issues of IG have highlighted in several works listed above. Likewise considering the geometry of the dataset under a Riemannian framework and integrating it into IG has been covered. Whilst the energy based approach is interesting the computational costs are not justified to promote the use of GIG over other efficient techniques.

**Claims And Evidence:**

No

**Claims Explanation:**

1) Little comparison with work which significantly overlaps with GIG

2) Limited datasets considered.

3) Insufficient metrics, thresholds, and base-points considered.

**Requested Changes:**

1) Address the significant related work as described above.
2) Increase the scope of the experimental validation in both datasets and metrics used.
3) Investigate the curvature of the data-manifolds and discuss the impact. See my above comment about Shao et al.'s paper and the implications of zero curvature for your work.

---

### Note · Authors · 2026-04-15

**Comment:**

Dear Action Editor,

Thank you for coordinating the review of our submission "Geodesic Integrated Gradients" (Paper 7664), and thank you to the reviewers for their detailed and constructive feedback.

We appreciate the engagement with both the theoretical and empirical aspects of the work. The reviewers raise fair points regarding the scope of experimental validation and the positioning of the Riemannian framing relative to the practical method. We do not currently have the resources to address the experimental gaps to the standard the reviewers rightly expect, and we believe the paper would benefit from a more careful reframing before resubmission.

We would therefore like to withdraw the paper.

Thank you again for the time and care invested in the reviews.

Best regards, Authors

**Withdrawal Confirmation:**

I have read and agree with the venue's withdrawal policy on behalf of myself and my co-authors.